

# The comparison of molecular and morphology-based phylogenies of trichaline net-winged beetles (Coleoptera: Lycidae: Metriorrhynchini) with description of a new subgenus

Matej Bocek and Ladislav Bocak

Department of Zoology, Faculty of Science, Palacky University, Olomouc, Czech Republic

## ABSTRACT

Separate morphological and molecular phylogenetic analyses are presented and the classification of trichaline net-winged beetles is revised. The clade, earlier given a subfamily, tribe or subtribe rank, is a terminal lineage in Metriorrhynchina and contains *Diatrichalus* Kleine, 1926, *Eniclases* Waterhouse, 1879, *Flabellotrichalus* Pic, 1921, *Lobatang* Bocak, 1998, *Microtrichalus* Pic, 1921, *Schizotrichalus* Kleine, 1926, and *Trichalus* Waterhouse, 1877. *Maibrius* subgen. nov. is proposed in *Flabellotrichalus* with the type-species *Flabellotrichalus* (*Maibrius*) *horaki* sp. nov. Unlike previous studies, *Lobatang* is included in the trichaline clade. Further, *Spinotrichalus* Kazantsev, 2010, stat. nov. is down-ranked to the subgenus in *Lobatang* Bocak, 1998 and a new combination, *Lobatang* (*Spinotrichalus*) *telnovi* (*Kazantsev, 2010*) comb. nov., is proposed. The morphology does not provide a sufficient support for robust phylogeny due to the intrageneric variability of most phenotypic traits and the limited number of characters supporting deep relationships. Most morphological generic diagnoses must be based on the shape of male genitalia. Other characters, such as the shapes of pronotum and antennae are commonly variable within genera. The fronto-lateral pronotal ridges of *Eniclases* + *Schizotrichalus* resemble the ancestral condition in Metriorrhynchini and they re-evolved in the terminal clade and do not indicate the early split of *Eniclases* + *Schizotrichalus* from other trichaline genera. The evolution of morphological traits and the conflict in the morphological and molecular phylogenetic signal are discussed in details. We suggest that the general appearance is affected by the evolution of mimetic complexes, the patterns of elytral costae by their strengthening function, and the presence of flabellate antennae by their role in sexual communication. Then, similar phenotypic traits evolve in unrelated lineages. The results demonstrate that phylogenetic classification must be based on all available information because neither morphological traits nor DNA data robustly support all recovered relationships.

Corresponding author
Ladislav Bocak,
ladislav.bocak@upol.cz

## INTRODUCTION

Based on morphological uniqueness, the trichaline genera were given various family group ranks from the subfamily to subtribe (*Kleine, 1928*; *Kleine, 1933a*; *Bocak & Bocakova, 1990*; *Bocak, 2002*). The molecular analyses recovered these genera as a terminal lineage in the subtribe Metriorrhynchina and to remedy this, they lost their formal rank (*Sklenarova, Kubecek & Bocak, 2014*). Although most of them are easily recognizable by a single lanceolate pronotal areola and a shortened elytral costa 1 (*Kleine, 1928*), the limits of the trichaline clade were questioned once the morphology was studied in detail (*Bocak, 1998a*; *Bocak, 2002*). Based on the morphological cladistic analysis, *Leptotrichalus Kleine, 1925*, and *Lobatang Bocak, 1998a* were excluded and *Enylus Waterhouse, 1879*, which is now a part of *Synchonnus Waterhouse, 1879* (*Kusy, Sklenarova & Bocak, in press*), was recovered as a member of Trichalini (*Bocak, 2002*). *Sklenarova, Kubecek & Bocak (2014)* revised the classification of Metriorrhynchini, but only *Trichalus Waterhouse, 1877*, and *Microtrichalus Pic, 1921b* were included in their analyses.

The trichaline clade contains approximately 230 formally described species and these represent ~20% of Metriorrhynchina diversity. There are high numbers of undescribed taxa in the various regions, as shown by recent studies (*Bocak & Bocakova, 1991*; *Kazantsev, 2010*; *Bocek & Bocak, 2016*; *Bocek, 2017*; *Kusy, 2017*). The trichaline species are currently placed in seven genera: *Diatrichalus Kleine, 1926*, *Eniclases Waterhouse, 1879*, *Flabellotrichalus Pic, 1921b*, *Microtrichalus*, *Schizotrichalus Kleine, 1926*, *Trichalus*, and, as shown below, *Lobatang*. The high variability of traditionally used phenotypic characters, especially variable general appearance, modifications of elytral costae and diverse morphology of male antennae, led to the description of a large number of genera in this clade (*Kleine, 1926*; *Pic, 1921b*, *1923*, *1926*, *1930*).

The center of trichaline diversity is located in the wet areas of the Australian region: the eastern coast of Australia (40 spp.), New Guinea (131 spp.), and Wallacea (31 spp.). Only a low number of species reach the Oriental region, mainly the Philippines (nine spp.), and the Greater Sundas (22 spp.). Several Indo-Burman species reach as far as the south of the Palearctic region (*Kleine, 1933a*; *Bocak, 1998b*, *1999a*). The first Australian representatives were already described from specimens brought to Europe in the time of discovery expeditions to the Southern Seas (*Fabricius, 1775*; *Boisduval, 1835*). Further species were described in the 19th century, many in other metriorrhynchine genera (*Erichson, 1842*; *Blanchard, 1856*; *Kirsch, 1875*; *Macleay, 1886*, *1887*; *Fairmaire, 1877*; *Waterhouse, 1877*, *1878*, *1879*; *Bourgeois, 1900*). A. M. Lea, R. Kleine, and M. Pic described over 150 species mainly in 1920s and 1930s (*Lea, 1909*; *Kleine, 1925*, *1926*, *1930*, *1936*, *1939*; *Pic, 1921a*, *1921b*, *1923*, *1926*, *1930*). *Diatrichalus* and *Microtrichalus* were partly revised in a series of geographically restricted revisions (*Bocak & Bocakova, 1991*; *Bocak, 1998b*, *1999a*, *2000*, *2001*). Later, only a single genus, *Spinotrichalus*, and four trichaline species, were described by *Kazantsev (2010)*.

A growing amount of DNA data is currently available for the molecular phylogeny reconstruction of trichaline genera (*Sklenarova, Kubecek & Bocak, 2014*; *Bocek & Bocak, 2016*). The aim of this study is to use morphology and molecular phylogeny for the

delimitation of genera and build a hypothesis on their relationships. The generic classification should reflect the best supported phylogenetic hypothesis, include only the monophyletic taxa, and be stable. Simultaneously, the genera should also be reliably identified in practice by the evaluation of phenotypic traits (*Vences et al., 2013*), ideally in the field, or by using simple laboratory equipment. Therefore, we discuss in detail the phenotypic diversification of trichaline genera and the usefulness of various morphological characters for both, phylogenetic inference and diagnostic purposes.

## MATERIALS AND METHODS

### Sampling, laboratory procedures, and sequence handling

The trichaline net-winged beetles included in current molecular analyses are listed in Table 1. Most terminals in the dataset are identified to the genus level only due to the ambiguous alpha-taxonomy and a high proportion of undescribed species in the dataset. Total DNA was isolated from ethanol-preserved individuals using Wizard SV96 DNA purification system (Promega Inc., Madison, WI, USA). All samples were sequenced for three mtDNA markers: *rrnL* + tRNA-Leu + *nad1* (~800 bp), *cox1* + tRNA-Leu + *cox2* (~1,100 bp), and *nad5* + tRNAs (~1,210 bp; the fragments are further referred as *rrnL*, *cox1*, and *nad5*) using primers reported by *Bocak et al. (2008)* and *Sklenarova, Chesters & Bocak (2013)*. The chromatograms were edited using the Sequencher 4.9 software package (Gene Codes Corp., Ann Arbor, MI, USA). The newly reported sequences were submitted to GenBank under Accession Numbers MF288149–MF288557 and MF997538–MF997543 (Table 1). Altogether 21 taxa were chosen from previous publication as outgroups. These represent all known Metriorrhynchina major lineages as identified by *Bocak et al. (2008)*, *Sklenarova, Chesters & Bocak (2013)*, and *Sklenarova, Kubecek & Bocak (2014)*. We avoided inclusion of all known ~150 Metriorrhynchini species available in public databases, as we did not intend to repeat the thorough analysis of the Metriorrhynchini published earlier. Additionally, the high number of distantly related taxa may affect the relationships within ingroup and affect its internal topology as demonstrated by *Bocak et al. (2014)*.

All voucher specimens, including the type material, are deposited in the voucher collection of the Department of Zoology, Palacky University in Olomouc, Czech Republic (LMBC).

### Phylogenetic analyses of the molecular dataset

Each DNA fragment was separately aligned with MAFFT 7.017 plug-in (*Katoh & Standley, 2013*) in Geneious R7.1.9 (Biomatters Inc., Newark, NJ, USA) and G-Ins-i algorithm. The alignment of the protein-coding genes *cox1*, *cox2*, *nad1*, and *nad5* were checked by amino acid reading frames and manually corrected, if necessary. The concatenated supermatrix was partitioned using PartitionFinder2 for all fragments and codon positions when appropriate (*Lanfear et al., 2014*, *2016*). The following partitions and models were proposed for the maximum-likelihood (ML) and Bayesian analyses. The RAxML best partitioning scheme: 13 subsets; 1 = 1–617, 2 = 618–684, 1,592–1,651, 3 = 1,912–2,925\3, 685–808\3; 4 = 686–808\3, 1,913–2,925\3, 5 = 687–808\3, 6 = 809–1,591\3, 7 = 810–1,591\3, 8 = 811–1,591\3, 9 = 1,652–1,911\3, 10 = 1,653–1,911\3, 11 = 1,654–1,911\3, 12 = 1,914–

**Table 1 List of taxa.**

| Genus, species | Geographic origin | Voucher | Mitochondrial DNA fragments | | |
|---|---|---|---|---|---|
| | | UPOL | rrnL | cox1 | nad5 |
| **Outgroup** | | | | | |
| *Cautires* sp. | Malaysia, Pahang, Tanah Rata | 000088 | KC538654 | KC538268 | KC538460 |
| *Cautires* sp. | Sumatra, Jambi, Gn Tujuh | 000206 | KC538676 | KC538292 | KC538483 |
| *Cautires* sp. | Borneo, Tenggah, Muara Teweh | 000262 | KC538685 | KC538300 | KC538491 |
| *Cautires* sp | Borneo, Selatan, Loksado | 000342 | KC538695 | KC538310 | KC538501 |
| *Porrostoma* sp. | Australia, Queensland, Lamington | A00035 | KC538725 | KC538341 | KC538532 |
| *Porrostoma* sp. | Australia, Queensland, Lamington | A00042 | – | KC538348 | KC538539 |
| *Leptotrichalus* sp. | Java, Timor, Sodong | A00451 | MF288196 | MF288334 | MF288457 |
| *Metriorrhynchus* sp. | Sulawesi, Tenggah, Sabbang | 000011 | KC538629 | DQ144660 | DQ144686 |
| *M. lineatus* | Sumatra, South, Danau Ranau | 000009 | KC538628 | DQ904297 | DQ904259 |
| *M. lobatus* | Sulawesi, Tenggah, Pendolo | 000017 | KC538630 | DQ144662 | DQ144688 |
| *M. sericans* | Laos, Houa Phan, Phou Pan | A00381 | MF288191 | MF288329 | MF288452 |
| *Metriorrhynchus* sp. | Australia, Queensland, Lamington | A00043 | KC538732 | KC538349 | KC538540 |
| *Metriorrhynchus* sp. | Malaysia, Johor, Kota Tinggi | A00049 | KC538736 | KC538354 | KC538545 |
| *Metriorrhynchus* sp. | Australia, Queensland, Bunya Mts. | A00311 | MF288174 | MF288312 | MF288437 |
| *Metriorrhynchus* sp. | Australia, Queensland, Lamington | A00348 | MF288183 | MF288320 | MF288445 |
| *Metriorrhynchus* sp. | New Guinea, Biak, Korim | A00422 | MF288192 | MF288330 | MF288453 |
| *Metriorrhynchus* sp. | New Guinea, Papau, Yiwika | BM0104 | MF288227 | MF288351 | MF288487 |
| Metriorrhynchina sp. | New Guinea, West Papua, Maibri | BM0083 | MF997538 | MF997540 | MF997542 |
| Metriorrhynchina sp. | New Guinea, Papua, Yiwika | BM0109 | MF997539 | MF997541 | MF997543 |
| *Synchonnus* sp. | Australia, Queensland, Lamington | A00039 | KC538729 | KC538345 | KC538536 |
| *Wakarumbia* sp. | Sulawesi, Mamasa | MD0155 | KC538809 | KC538432 | KC538624 |
| **Ingroup** | | | | | |
| *Diatrichalus* sp. A | Sulawesi, Selatan, Mamasa | JB0774 | – | MF288416 | – |
| *Diatrichalus* sp. B | Malaysia, Kelantan, Kp. Raja | JB0829 | – | MF288417 | – |
| *D. xylobanoides* | New Guinea, Crater Mt., Haia | A00118 | – | MF288291 | MF288419 |
| *D. dilatatus* | New Guinea, Goroka, Gahavisuka | A00133 | MF288151 | – | MF288544 |
| *D. mancus* | Australia, Queensland, Pascoe River | A00298 | MF288172 | MF288311 | MF288436 |
| *D. manokwarensis* | New Guinea, West Papua, Maibri | BM0079 | MF288216 | MF288343 | MF288477 |
| *D. mindikensis* | New Guinea, Morobe, Mindik | A00184 | MF288160 | – | MF288427 |
| *D. robustus* | New Guinea, Papua, Elelim | BM0190 | MF288288 | MF288412 | MF288555 |
| *D. robustus* | New Guinea, Papau, Elelim | BM0191 | MF288289 | MF288413 | MF288556 |
| *D. sinuaticollis* | New Guinea, Papua, Bokondini | BM0114 | MF288233 | MF288357 | MF288550 |
| *Diatrichalus* sp. C | New Guinea, Papua, Yiwika | BM0113 | MF288232 | MF288356 | MF288492 |
| *Diatrichalus* sp. D | New Guinea, Papua, Tikapura | BM0127 | MF288245 | MF288369 | MF288504 |
| *Diatrichalus* sp. E | New Guinea, Papua, Elelim | BM0159 | MF288267 | MF288391 | MF288526 |
| *Diatrichalus* sp. F | New Guinea, Papua, Elelim | BM0192 | MF288290 | MF288414 | MF288557 |
| *Diatrichalus* sp. G | Australia, Queensland, Chilverton | A00208 | MF288163 | MF288302 | MF288546 |
| *Diatrichalus* sp. G | Australia, Queensland, Chilverton | A00237 | MF288167 | MF288306 | MF288547 |
| *Diatrichalus* sp. G | Australia, Queensland, Garradunga | A00308 | MF288173 | – | – |

| Genus, species | Geographic origin | Voucher | Mitochondrial DNA fragments | | |
| --- | --- | --- | --- | --- | --- |
| | | UPOL | *rrnL* | *cox1* | *nad5* |
| *Diatrichalus* sp. G | Australia, Queensland, Garradunga | A00337 | MF288181 | – | MF288548 |
| *Diatrichalus* sp. H | New Guinea, Papua, Tikapura | BM0189 | MF288287 | MF288411 | MF288554 |
| *Diatrichalus* sp. I | New Guinea, Goroka, Gahavisuka | A00131 | MF288150 | – | – |
| *Diatrichalus* sp. I | New Guinea, Goroka, Gahavisuka | A00156 | MF288154 | MF288295 | MF288545 |
| *Diatrichalus* sp. J | New Guinea, Papua, Tikapura | BM0188 | MF288286 | MF288410 | MF288553 |
| *Diatrichalus* sp. K | New Guinea, West Papua, Wasior | JB0772 | – | MF288415 | – |
| *D. tenimberensis* | Australia, Queensland, Claudie River | A00366 | MF288190 | MF288328 | MF288549 |
| *Eniclases apertus* | New Guinea, Papua, Sentani | BM0018 | MF288201 | KT265155 | MF288462 |
| *E. bicolor* | New Guinea, Papua, Elelim | BM0045 | MF288204 | KT265166 | MF288465 |
| *E. bokondinensis* | New Guinea, Papua, Bokondini | BM0094 | MF288222 | KT265153 | MF288482 |
| *E. brancuccii* | New Guinea, Papua, Sentani | BM0005 | MF288199 | KT265118 | MF288460 |
| *E. divaricatus* | New Guinea, Papua, Sentani | BM0001 | MF288197 | KT265092 | MF288458 |
| *E. divaricatus* | New Guinea, Papua, Elelim | BM0057 | MF288207 | KT265098 | MF288468 |
| *E. elelimensis* | New Guinea, Papua, Elelim | BM0051 | MF288206 | KT265149 | MF288467 |
| *E. infuscatus* | New Guinea, Papua, Elelim | BM0050 | MF288205 | KT265169 | MF288466 |
| *E. niger* | New Guinea, Papua, Bokondini | BM0033 | MF288202 | KT265111 | MF288463 |
| *E. pseudoluteolus* | New Guinea, West Papua, Maibri | BM0084 | MF288219 | KT265171 | MF288480 |
| *E. similis* | New Guinea, Papua, Sentani | BM0003 | MF288198 | KT265099 | MF288459 |
| *Eniclases* sp. A | New Guinea, Papua, Bokondini | BM0093 | MF288221 | KT265163 | MF288481 |
| *E. tikapurensis* | New Guinea, Papua, Yiwika | BM0039 | MF288203 | KT265157 | MF288464 |
| *E. variabilis* | New Guinea, Papua, Sentani | BM0008 | MF288200 | KT265122 | MF288461 |
| *Flabellotrichalus* sp. A | New Guinea, Crater Mt., Haia | A00170 | MF288157 | MF288298 | MF288425 |
| *Flabellotrichalus* sp. B | New Guinea, Pindiu, Mongi | A00180 | MF288159 | MF288300 | MF288426 |
| *Flabellotrichalus* sp. C | New Guinea, Papua, Yiwika | BM0103 | MF288226 | MF288350 | MF288486 |
| *Flabellotrichalus* sp. C | New Guinea, Papua, Yiwika | BM0110 | MF288230 | MF288354 | MF288490 |
| *Flabellotrichalus* sp. C | New Guinea, Papua, Yiwika | BM0111 | MF288231 | MF288355 | MF288491 |
| *Flabellotrichalus* sp. D | New Guinea, Pt. Moresby, Kailaki | A00149 | MF288153 | MF288294 | MF288422 |
| *Flabellotrichalus* sp. D | New Guinea, Papua, Elelim | BM0148 | MF288257 | MF288381 | MF288516 |
| *Flabellotrichalus* sp. D | New Guinea, Papua, Elelim | BM0149 | MF288258 | MF288382 | MF288517 |
| *Flabellotrichalus* sp. D | New Guinea, Papua, Elelim | BM0150 | MF288259 | MF288383 | MF288518 |
| *Flabellotrichalus* sp. E | New Guinea, Crater Mt., Haia | A00172 | MF288158 | MF288299 | – |
| *Flabellotrichalus* sp. F | New Guinea, Crater Mt., Haia | A00125 | MF288149 | MF288292 | MF288420 |
| *Flabellotrichalus* sp. F | New Guinea, Crater Mt., Haia | A00162 | MF288155 | MF288296 | MF288423 |
| *Flabellotrichalus* sp. F | New Guinea, Crater Mt., Haia | A00169 | MF288156 | MF288297 | MF288424 |
| *Flabellotrichalus* sp. G | Australia, Queensland, Chilverton | A00211 | MF288165 | MF288304 | MF288430 |
| *Flabellotrichalus* sp. H | New Guinea, Papua, Yiwika | BM0105 | MF288228 | MF288352 | MF288488 |
| *Flabellotrichalus* sp. I | New Guinea, Papua, Elelim | BM0151 | MF288260 | MF288384 | MF288519 |
| *F. (Maibrius) horaki* | New Guinea, West Papua, Maibri | BM0082 | MF288218 | MF288345 | MF288479 |
| *Lobatang* sp. A | New Guinea, Papua, Sentani | BM0162 | MF288269 | MF288393 | MF288528 |
| *Lobatang* sp. A | New Guinea, Papua, Sentani | BM0168 | MF288274 | MF288398 | MF288533 |

(Continued)

| Genus, species | Geographic origin | Voucher | Mitochondrial DNA fragments | | |
|---|---|---|---|---|---|
| | | UPOL | rrnL | cox1 | nad5 |
| *Lobatang* sp. B | Australia, Queensland, Claudie River | A00363 | MF288187 | MF288325 | MF288450 |
| *Lobatang* sp. B | Australia, Queensland, Claudie River | A00365 | MF288189 | MF288327 | – |
| *Lobatang* sp. C | Moluccas, Buru isl., Remaja Mt. | BM0071 | MF288208 | MF288335 | MF288469 |
| *Lobatang* sp. C | Moluccas, Buru isl., Remaja Mt. | BM0072 | MF288209 | MF288336 | MF288470 |
| *Lobatang* sp. C | Moluccas, Buru isl., Remaja Mt. | BM0073 | MF288210 | MF288337 | MF288471 |
| *Lobatang* sp. | Moluccas, Buru isl., Remaja Mt. | BM0074 | MF288211 | MF288338 | MF288472 |
| *Lobatang* sp. D | New Guinea, West Papua, Maibri | BM0075 | MF288212 | MF288339 | MF288473 |
| *Lobatang* sp. D | New Guinea, West Papua, Maibri | BM0076 | MF288213 | MF288340 | MF288474 |
| *Lobatang* sp. D | New Guinea, Papua, Elelim | BM0145 | MF288254 | MF288378 | MF288513 |
| *Lobatang* sp. D | New Guinea, Papua, Elelim | BM0146 | MF288255 | MF288379 | MF288514 |
| *Lobatang* sp. D | New Guinea, Papua, Sentani | BM0165 | MF288271 | MF288395 | MF288530 |
| *Lobatang* sp. D | New Guinea, Papua, Sentani | BM0166 | MF288272 | MF288396 | MF288531 |
| *Microtrichalus* sp. A | New Guinea, Papua, Sentani | BM0175 | MF288277 | MF288401 | MF288551 |
| *Microtrichalus* sp. A | New Guinea, Papua, Sentani | BM0180 | MF288281 | MF288405 | MF288552 |
| *Microtrichalus* sp. B | New Guinea, Papua, Sentani | BM0178 | MF288279 | MF288403 | MF288537 |
| *Microtrichalus* sp. B | New Guinea, Papua, Sentani | BM0179 | MF288280 | MF288404 | MF288538 |
| *Microtrichalus* sp. C | Australia, Queensland, Claudie River | A00356 | – | MF288322 | MF288447 |
| *Microtrichalus* sp. C | Australia, Queensland, Claudie River | A00364 | MF288188 | MF288326 | MF288451 |
| *Microtrichalus* sp. D | New Guinea, Papua, Elelim | BM0158 | MF288266 | MF288390 | MF288525 |
| *Microtrichalus* sp. E | New Guinea, Papua, Tikapura | BM0134 | MF288247 | MF288371 | MF288506 |
| *Microtrichalus* sp. F | New Guinea, Papua, Bokondini | BM0117 | MF288236 | MF288360 | MF288495 |
| *Microtrichalus* sp. F | New Guinea, Papua, Tikapura | BM0135 | MF288248 | MF288372 | MF288507 |
| *Microtrichalus* sp. G | New Guinea, Papua, Yiwika | BM0102 | MF288225 | MF288349 | MF288485 |
| *Microtrichalus* sp. G | New Guinea, Papua, Tikapura | BM0126 | MF288244 | MF288368 | MF288503 |
| *Microtrichalus* sp. H | New Guinea, West Papua, Maibri | BM0077 | MF288214 | MF288341 | MF288475 |
| *Microtrichalus* sp. H | New Guinea, West Papua, Maibri | BM0085 | MF288220 | MF288346 | – |
| *Microtrichalus* sp. I | New Guinea, Papua, Bokondini | BM0122 | MF288241 | MF288365 | MF288500 |
| *Microtrichalus* sp. I | New Guinea, Papua, Bokondini | BM0123 | MF288242 | MF288366 | MF288501 |
| *Microtrichalus* sp. I | New Guinea, Papua, Elelim | BM0152 | MF288261 | MF288385 | MF288520 |
| *Microtrichalus* sp. I | New Guinea, Papua, Elelim | BM0153 | MF288262 | MF288386 | MF288521 |
| *Microtrichalus* sp. J | Australia, Queensland, Chilverton | A00239 | MF288168 | MF288307 | MF288432 |
| *Microtrichalus* sp. J | Australia, Queensland, Chilverton | A00243 | MF288169 | MF288308 | MF288433 |
| *Microtrichalus* sp. K | New Guinea, Papua, Sentani | BM0160 | MF288268 | MF288392 | MF288527 |
| *Microtrichalus* sp. K | New Guinea, Papua, Sentani | BM0164 | MF288270 | MF288394 | MF288529 |
| *Microtrichalus* sp. K | New Guinea, Papua, Sentani | BM0167 | MF288273 | MF288397 | MF288532 |
| *Microtrichalus* sp. K | New Guinea, Papua, Sentani | BM0169 | MF288275 | MF288399 | MF288534 |
| *Microtrichalus* sp. L | New Guinea, Papua, Elelim | BM0147 | MF288256 | MF288380 | MF288515 |
| *Microtrichalus* sp. M | Australia, Queensland, Claudie River | A00353 | MF288184 | MF288321 | MF288446 |
| *Microtrichalus* sp. N | New Guinea, Papua, Bokondini | BM0119 | MF288238 | MF288362 | MF288497 |
| *Microtrichalus* sp. O | New Guinea, Papua, Napua | BM0185 | MF288283 | MF288407 | MF288540 |

| Genus, species | Geographic origin | Voucher | Mitochondrial DNA fragments | | |
|---|---|---|---|---|---|
| | | UPOL | *rrnL* | *cox1* | *nad5* |
| *Microtrichalus* sp. O | New Guinea, Papua, Tikapura | BM0141 | MF288253 | MF288377 | MF288512 |
| *Microtrichalus* sp. P | Australia, Queensland, Mt. Molloy | 000375 | KC538702 | KC538315 | KC538506 |
| *Microtrichalus* sp. P | Australia, Queensland, Pascoe River | A00314 | MF288176 | MF288314 | MF288439 |
| *Microtrichalus* sp. P | Australia, Queensland, Pascoe River | A00315 | MF288177 | MF288315 | MF288440 |
| *Microtrichalus* sp. P | Australia, Queensland, Pascoe River | A00316 | MF288178 | MF288316 | MF288441 |
| *Microtrichalus* sp. Q | Australia, Queensland, Chilverton | A00210 | MF288164 | MF288303 | MF288429 |
| *Microtrichalus* sp. R | New Guinea, Papua, Sentani | BM0183 | MF288282 | MF288406 | MF288539 |
| *Microtrichalus* sp. S | New Guinea, Papua, Bokondini | BM0120 | MF288239 | MF288363 | MF288498 |
| *Microtrichalus* sp. T | Australia, Queensland, Chilverton | A00206 | MF288162 | MF288301 | – |
| *Microtrichalus* sp. T | Australia, Queensland, Chilverton | A00235 | MF288166 | MF288305 | MF288431 |
| *Microtrichalus* sp. T | Australia, Queensland, Duintrea | A00192 | MF288161 | – | MF288428 |
| *Microtrichalus* sp. U | New Guinea, Papua, Yiwika | BM0108 | MF288229 | MF288353 | MF288489 |
| *Microtrichalus* sp. V | New Guinea, Papua, Bokondini | BM0115 | MF288234 | MF288358 | MF288493 |
| *Microtrichalus* sp. W | New Guinea, Goroka, Gahavisuka | A00139 | MF288152 | MF288293 | MF288421 |
| *Microtrichalus* sp. X | New Guinea, Papua, Yiwika | BM0100 | MF288223 | MF288347 | MF288483 |
| *Microtrichalus* sp. X | New Guinea, Papua, Napua | BM0186 | MF288284 | MF288408 | MF288541 |
| *Microtrichalus* sp. Y | Australia, Queensland, Claudie River | A00270 | MF288170 | MF288309 | MF288434 |
| *Microtrichalus* sp. Y | Australia, Queensland, Claudie River | A00357 | MF288185 | MF288323 | MF288448 |
| *Microtrichalus* sp. Y | Australia, Queensland, Claudie River | A00362 | MF288186 | MF288324 | MF288449 |
| *Microtrichalus* sp. Z | New Guinea, Papua, Bokondini | BM0121 | MF288240 | MF288364 | MF288499 |
| *Microtrichalus* sp. Z | New Guinea, Papua, Bokondini | BM0124 | MF288243 | MF288367 | MF288502 |
| *Microtrichalus* sp. Z | New Guinea, Papua, Sentani | BM0177 | MF288278 | MF288402 | MF288536 |
| *Microtrichalus* sp. AA | Borneo, Sabah, Poring | MK0852 | – | MF288418 | MF288543 |
| *Microtrichalus* sp. AB | New Guinea, Papua, Bokondini | BM0116 | MF288235 | MF288359 | MF288494 |
| *Microtrichalus* sp. AB | New Guinea, Papua, Bokondini | BM0118 | MF288237 | MF288361 | MF288496 |
| *Microtrichalus* sp. AC | New Guinea, West Papua, Maibri | BM0081 | MF288217 | MF288344 | MF288478 |
| *Microtrichalus* sp. AD | New Guinea, Papua, Elelim | BM0154 | MF288263 | MF288387 | MF288522 |
| *Microtrichalus* sp. AD | New Guinea, Papua, Elelim | BM0156 | MF288264 | MF288388 | MF288523 |
| *Microtrichalus* sp. AD | New Guinea, Papua, Elelim | BM0157 | MF288265 | MF288389 | MF288524 |
| *Trichalus* sp. A | Australia, Queensland, Lamington | A00032 | KC538722 | KC538339 | KC538529 |
| *Trichalus* sp. B | Australia, Queensland, Tinarooo | A00312 | MF288175 | MF288313 | MF288438 |
| *Trichalus* sp. B | Australia, Queensland, Fletcher Creek | A00320 | MF288179 | MF288317 | MF288442 |
| *Trichalus* sp. B | Australia, Queensland, Mt. Garnet | A00342 | MF288182 | MF288319 | MF288444 |
| *Trichalus* sp. C | Australia, Queensland, Garradunga | A00336 | MF288180 | MF288318 | MF288443 |
| *Trichalus* sp. D | Australia, Queensland, Fletcher Creek | A00287 | MF288171 | MF288310 | MF288435 |
| *T. communis* | Malaysia, Kelantan, Gua Musang | A00425 | MF288193 | MF288331 | MF288454 |
| *T. communis* | Malaysia, Kelantan, Gua Musang | A00426 | MF288194 | MF288332 | MF288455 |
| *Trichalus* sp. E | New Guinea, West Papua, Maibri | BM0078 | MF288215 | MF288342 | MF288476 |
| *Trichalus* sp. F | New Guinea, Papua, Sentani | BM0174 | MF288276 | MF288400 | MF288535 |
| *Trichalus* sp. G | New Guinea, Papua, Napua | BM0187 | MF288285 | MF288409 | MF288542 |

(Continued)

| Genus, species | Geographic origin | Voucher | Mitochondrial DNA fragments | | |
|---|---|---|---|---|---|
| | | UPOL | *rrnL* | *cox1* | *nad5* |
| *Trichalus* sp. H | New Guinea, Papua, Tikapura | BM0136 | MF288249 | MF288373 | MF288508 |
| *Trichalus* sp. H | New Guinea, Papua, Tikapura | BM0140 | MF288252 | MF288376 | MF288511 |
| *Trichalus* sp. I | New Guinea, Papua, Tikapura | BM0133 | MF288246 | MF288370 | MF288505 |
| *Trichalus* sp. J | New Guinea, Papua, Tikapura | BM0138 | MF288250 | MF288374 | MF288509 |
| *Trichalus* sp. J | New Guinea, Papua, Yiwika | BM0101 | MF288224 | MF288348 | MF288484 |
| *Trichalus* sp. J | New Guinea, Papua, Tikapura | BM0139 | MF288251 | MF288375 | MF288510 |

**Note:**
The list of terminals in the molecular phylogenetic analyses, with voucher and GenBank accession numbers.

2,925\3, 13 = 2,926–3,184. The model GTR+I+G was proposed for subsets 1–9 and 13 and GTR+G for subsets 10–12. The model GTR+I+G was applied for all subsets in the maximum-likelihood analyses as RAxML allows for only a single model of rate heterogeneity in partitioned analyses. I.e., we assigned GTR+I+G as the model providing the most accurate estimation of the DNA evolution (*Stamatakis, 2014*; *Lanfear et al., 2014*, *2016*). The position cited refers to those in the supermatrix provided as the File S1, i.e., the aligned DNA dataset used for the ML analysis. The BI best partitioning scheme: 14 subsets; 1 = 1–617, 2 = 618–684, 1,592–1,651, 3 = 1,912–2,925\3, 685–808\3, 4 = 686–808\3, 5 = 687–808\3, 6 = 809–1,591\3, 7 = 810–1,591\3, 8 = 811–1,591\3, 9 = 1,652–1,911\3, 10 = 1,653–1,911\3, 11 = 1,654–1,911\3, 12 = 1,913–2,925\3, 13 = 1,914–2,925\3, 14 = 2,926–3,184. The model GTR+I+G was proposed for subsets 1–9, 13–14 and GTR+G for subsets 10–12. The models were applied in the BI analysis as proposed by PartitionFinder2. The position refers to the alignment provided in the File S1 as above.

We used the ML criterion and Bayesian interference (BI) for phylogenetic analyses of the partitioned supermatrix (File S1). The ML searches were conducted in RAxML 8.2.10 (*Stamatakis, 2014*) on the CIPRES cluster (*Miller, Pfeiffer & Schwartz, 2010*) with the partitions described above and the GTR+I+G model identified using PartitionFinder2 as described above. Additionally, we analyzed the dataset with the partition by genes and protein coding positions when appropriate and the GTR+I+G model identified by jModelTest 2.1.7 (*Darriba et al., 2012*). Bootstrap support values were calculated in both analyses from 1,000 pseudoreplicates using the GTR+I+G model proposed by PartitionFinder2 or using the GTRCAT model which enables a time-effective and still sufficiently precise estimation of the bootstrap support in the analysis using partitions by genes (*Stamatakis, 2014*). The BI analysis was run in MrBayes 3.2.6 (*Ronquist et al., 2012*) on the CIPRES cluster under the best partitioning scheme suggested by PartitionFinder2 (*Lanfear et al., 2014*, *2016*; see above) for $6 \times 10^7$ generations, sampling a single tree every 1,200 generations. The first 5,000 trees were discarded as burn-in after the identification of the stationary phase and the effective sample size in Tracer 1.6 (*Rambaut et al., 2014*). The same analysis was run with gene partitions and GTR+I+G model as proposed by jModelTest 2.1.7 (*Darriba et al., 2012*). Posterior probabilities (PP) were calculated from the post-burn-in trees and mapped on the maximum credibility tree. Both trees produced
**Table 2 Morphological dataset.**

| Characters<br>Taxa | 0000000001111111111222222222<br>1234567890123456789012345678 |
|---|---|
| *Metriorrhynchus* | 00000-0000000000000000000000 |
| *Kassemia* | 001001000000110000000000010 |
| *Synchonnus* | 011000000011000001000000010 |
| *Diatrichalus* | 01101000011101-0010000000011 |
| *Leptotrichalus* | 00010010001101000000000100000 |
| *Lobatang* | 000000000011010100001010000-0 |
| *Schizotrichalus* | 01010000101001-0100100010000 |
| *Eniclases* | 11010-0010101100100100010000 |
| *Flabellotrichalus* | 110101010011010010000101010000 |
| *Trichalus* | 110000000111010010101-00010000 |
| *Microtrichalus* | 110100000111010010101000011100 |

Note:
   The description of character states is provided in the text.

by ML and BI analyses were rooted by *Cautires Waterhouse, 1879* (the type genus of the sister subtribe Cautirina, see *Bocak et al., 2008*; *Sklenarova, Chesters & Bocak, 2013*; *Sklenarova, Kubecek & Bocak, 2014*). The rooting forces Metriorrhynchina to be a clade, but we do not force trichaline genera to be monophyletic and their monophyly can be rigorously re-tested by the current analysis. All trees were visualized in FigTree 1.4.2 (http://tree.bio.ed.ac.uk/software/figtree) and edited in a graphic software.

## Morphological phylogeny

Adult semaphoronts were used for morphological descriptions. Male and female genitalia were relaxed and cleared in hot 10% KOH, dissected and stained by chlorazol black when needed. All photographs were taken using a camera on an Olympus SZX-16 binocular microscope. The morphological measurements were taken with the ocular scale.

The characters from earlier published morphological datasets (*Bocak, 1998a*, *2002*) and the newly identified characters (*Kazantsev, 2010*) were compiled in a single dataset of 11 taxa and 28 characters (Table 2; File S2). *Metriorrhynchus* was considered as an outgroup when the tree was rooted. The characters in the trichaline clade were polarized by the outgroup criterion. The autapomorphies of genera are based on inspection of all available taxa classified in the respective genus and they are included in the analysis to map their distribution. These characters do not affect the topology. The following characters were coded for all genera of the trichaline clade and taxa representing non-trichaline Metriorrhynchina:

1. **Shape of external mandibular margin in ventral view:** (0) nearly straight; (1) concave.
2. **Shape of mandibles:** (0) slightly curved or sickle-shaped; (1) apical part curved in right angle.
3. **Shape of mandibular incisor:** (0) inner margin twice broken; (1) inner margin continuously curved.

4. **Shape of apical maxillary palpomere:** (0) securiform; (1) parallel-sided, more or less obliquely cut at apex.

5. **Presence of sensillae at apex of terminal palpomere:** (0) absent; (1) present.

6. **Shape of male antennae:** (0) male antennae filiform to serrate; (1) antennomeres 3–10 flabellate.

7. **Shape of pronotum:** (0) approximately as long as wide; (1) much longer than wide.

8. **Pubescence of pronotum:** (0) whole pronotum with pubescence of the same type and density; (1) apparently denser and longer pubescence at lateral and frontal margins.

9. **Strength of hind margin of metascutellum:** (0) hind margin of metascutellum simple; (1) bent, strengthened.

10. **Shape of hind margin of metascutellum and presence of the metascutellar keel:** (0) hind margin of metascutellum straight, without keel; (1) emarginate, with keel.

11. **Arrangement of pronotal carinae:** (0) seven pronotal areolae; (1) less than seven pronotal areolae.

12. **Number of pronotal areolae:** (0) at least five areolae or at least vestiges of frontal and postero-lateral keels present; (1) only a lanceolate median areola present.

13. **Strengthened pronotal longitudinal carinae:** (0) absent; (1) present.

14. **The number of fully developed elytral primary costae in middle part of elytron:** (0) four primary costae; (1) three primary costae.

15. **Secondary elytral costae:** (0) secondary costae present; (1) absent.

16. **Split tarsal claws:** (0) no; (1) yes.

17. **Shape of apical part of phallus:** (0) wider or as wide as its middle part, only in apical part open, if apical part slender, then well-sclerotized and internal sac widely exposed; (1) apical part of phallus slender, with cup-shaped apex, only dorsal part sclerotized.

18. **Phallus short, robust, sometimes with a ventral process:** (0) no; (1) yes.

19. **Sickle-shaped thorns at base of internal sac:** (0) absent; (1) present.

20. **Single keel in dorsal part of phallus:** (0) absent; (1) present.

21. **Internal sac:** (0) membranous or with sclerotized sclerites in apical part; (1) rod-shaped at least in the basal part.

22. **Internal sac with y-shaped base:** (0) no; (1) yes.

23. **Shape of valvifers:** (0) valvifers long, slender; (1) valvifers short, fused with coxites.

24. **Attachment of lateral vaginal glands:** (0) laterally; (1) dorsally.

25. **Lateral pockets on vagina:** (0) absent; (1) present.

26. **Unpaired slim vaginal gland:** (0) absent; (1) present.

27. **Length of spermatheca:** (0) relatively short, lemon-like; (1) long, slender.

28. **Structure of the basal part of the spermathecal duct:** (0) slim; (1) robust.

The maximum parsimony (MP) analysis was performed using PAUP* 4.0 (*Swofford, 2002*). Heuristic searches were conducted with 1,000 repetitions and random stepwise additions; all characters were unordered and equally weighted and polymorphic

characters were treated as "missing" data. The level of confidence in each node of the MP trees was assessed using bootstrapping based on 1,000 pseudoreplicates, each analysis with 100 random additions. Further, we estimated morphology-based phylogenetic relationships using Bayesian inference as implemented in BEAST 2 (*Bouckaert et al., 2013*). The analysis was conducted using Lewis MK substitution model, a lognormal relaxed clock model, and a birth–death tree prior. The number of generation was set to $10^7$ and sampling frequency every 1,000 generation. We used Tracer 1.6 (*Rambaut et al., 2014*) to confirm convergence, and based on this, we discarded the first 25% of generations as burn-in. We used the program TreeAnnotator 2.4.5 (*Bouckaert et al., 2013*) to produce maximum clade credibility tree with PP.

The electronic version of this article in portable document format (PDF) will represent a published work according to the International Commission on Zoological Nomenclature (ICZN), and hence the new names contained in the electronic version are effectively published under that Code from the electronic edition alone. This published work and the nomenclatural acts it contains have been registered in ZooBank, the online registration system for the ICZN. The ZooBank LSIDs (Life Science Identifiers) can be resolved and the associated information viewed through any standard web browser by appending the LSID to the prefix http://zoobank.org/. The LSID for this publication is: urn:lsid:zoobank.org:pub:BCDB57BC-DF3E-42A8-AB6D-2DCAB44799F3. The online version of this work is archived and available from the following digital repositories: PeerJ, PubMed Central and CLOCKSS.

# RESULTS

## Molecular analysis

The molecular dataset contained 143 ingroup terminals representing 86 species from the whole range of the trichaline clade. Three markers were sequenced: *rrnL* mtDNA (137 ingroup samples), *cox1–3′* end of mtDNA (137 samples), and *nad5* mtDNA (134 samples). The concatenated dataset consisted of 3,184 homologous positions: the alignments of the *rrnL*, *cox1*, and *nad5* fragments contained 808, 1,103, and 1,273 homologous base pairs, respectively. The phylogenetic trees inferred from the MAFFT alignment using the ML criterion and Bayesian inference were well-resolved and suggested similar relationships. The differences in the applied partitions and models proposed by PartitionFinder2 and jModelTest 2.1.7 did not have any effect on the ML topology and the bootstrap support values inferred in both analyses were highly similar and the topology is shown in Fig. 1A and Fig. S1. The differences reached up to 2% and can be explained by the stochastic character of bootstrap analyses. The results of analyses based on the jModelTest partitions and models are not shown and they are not discussed further. The BI topology differs only slightly in the outgroup and internal topology of the *Microtrichalus* clade (Fig. S1). However, ambiguities in hypothesized relationships within *Microtrichalus* were expected as all ML and BI analyses recovered low BS and PP values for most internal relationships (Fig. 1A; Fig. S1). The differences between analyses were limited to rearrangements in *Microtrichalus* clade and did not include relationships among genera (Fig. 1A). The trichaline clade was regularly recovered although only with an ambiguous support

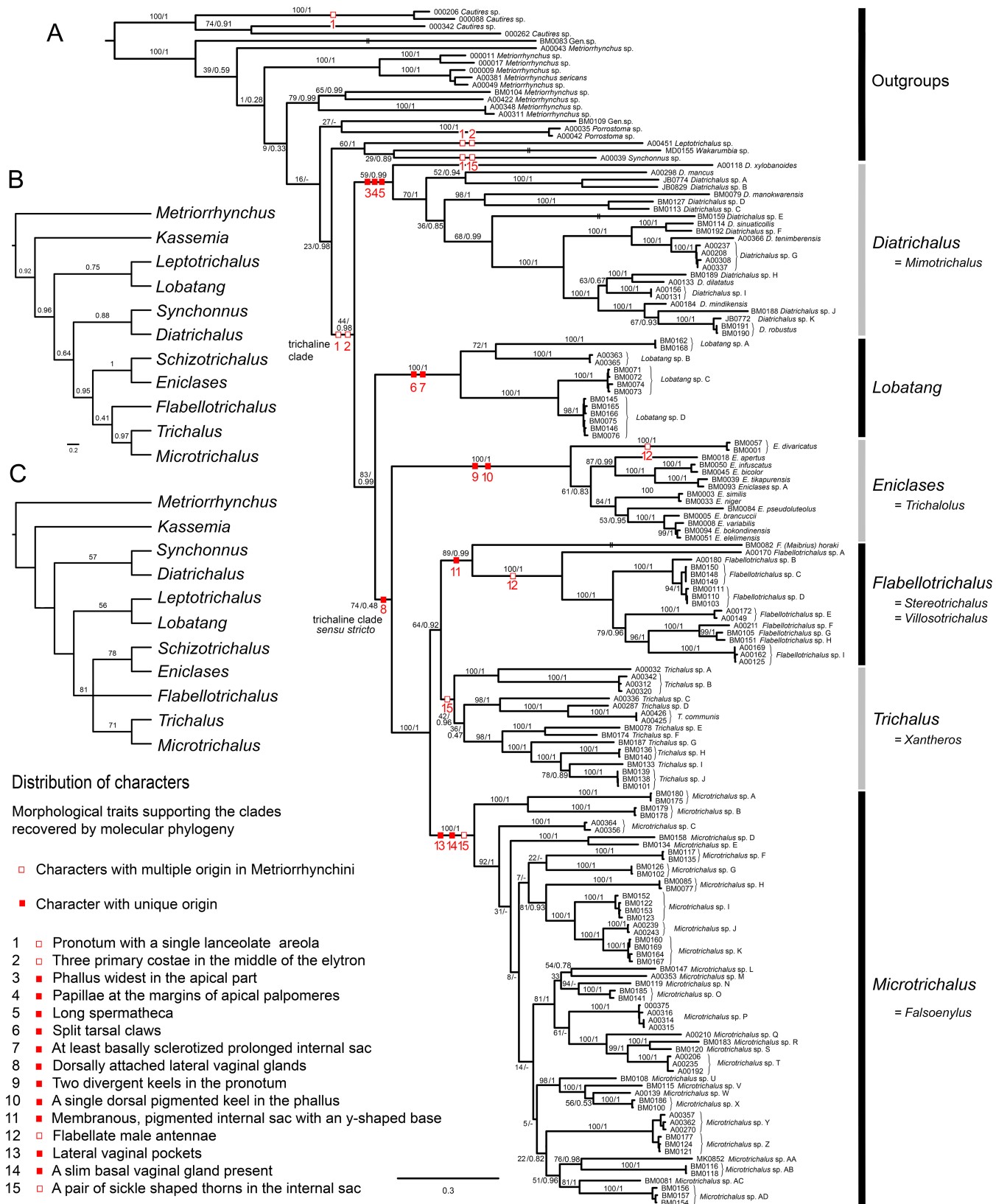

Distribution of characters

Morphological traits supporting the clades
recovered by molecular phylogeny

☐  Characters with multiple origin in Metriorrhynchini

■  Character with unique origin

| | | |
|---|---|---|
| 1 | ☐ | Pronotum with a single lanceolate areola |
| 2 | ☐ | Three primary costae in the middle of the elytron |
| 3 | ■ | Phallus widest in the apical part |
| 4 | ■ | Papillae at the margins of apical palpomeres |
| 5 | ■ | Long spermatheca |
| 6 | ■ | Split tarsal claws |
| 7 | ■ | At least basally sclerotized prolonged internal sac |
| 8 | ■ | Dorsally attached lateral vaginal glands |
| 9 | ■ | Two divergent keels in the pronotum |
| 10 | ■ | A single dorsal pigmented keel in the phallus |
| 11 | ■ | Membranous, pigmented internal sac with an y-shaped base |
| 12 | ☐ | Flabellate male antennae |
| 13 | ■ | Lateral vaginal pockets |
| 14 | ■ | A slim basal vaginal gland present |
| 15 | ☐ | A pair of sickle shaped thorns in the internal sac |

**PeerJ** _________________________________________________

**Figure 1 Phylogenetic hypotheses.** (A) Molecular phylogenetic reconstruction of trichaline relationships using maximum-likelihood; (B) Bayesian phylogenetic reconstruction of trichaline morphological relationships, the maximum clade credibility tree with posterior probabilities mapped; (C) phylogenetic reconstruction of trichaline relationships inferred from morphology using the parsimony criterion. The topologies in B and C were inferred from morphological dataset shown in Table 1. The numbers at branches show bootstrap support values (A, values before slash and C) and posterior probabilities (A, values after slash, B). Only values over 50% shown in (C). Voucher numbers at branch tips identify the samples listed in Table 1.

(BS 44%, PP 0.98). *Diatrichalus* marked the deepest node, followed by *Lobatang* and a clade of *Eniclases, Trichalus, Flabellotrichalus,* and *Microtrichalus,* further designated as the trichaline clade *sensu stricto. Schizotrichalus* was unavailable for molecular analyses. The genus-rank clades obtained mostly robust support >90% and regularly PP ~1.0, except *Diatrichalus* (BS 59%, PP 0.99) and *Trichalus* (BS 42%, PP 0.96). The relationships among these deep nodes remain poorly supported. The sister clade of trichaline genera contains *Leptotrichalus, Synchonnus,* and *Wakarumbia* Bocak, 1999b.

## Morphological analysis

The morphological analyses did not support the monophyly of the DNA-based trichaline clade (Figs. 1B and 1C). The relationships of *Schizotrichalus, Eniclases, Flabellotrichalus, Microtrichalus,* and *Trichalus* were satisfactorily resolved only by the BI analysis (Fig. 1B; Fig. S2), but the MP analysis recovered three equally parsimonious trees ($L = 38$, $CI = 0.737$, $RI = 0.714$). Their strict consensus and one of the most parsimonious trees were unresolved (Fig. 1C). The deeper relationships were poorly supported. The only synapomorphy which confirms the monophyly of the (*Schizotrichalus, Eniclases*), *Flabellotrichalus,* (*Microtrichalus, Trichalus*) clade are the dorsally attached lateral vaginal glands (Figs. 1A and 6Q). The presence of thorns in the internal sac suggests relationships of *Trichalus* and *Microtrichalus* and the pigmented keel supports relationships of *Eniclases + Schizotrichalus.* All discussed character states, including apomorphies which support individual genera, are mapped on the molecular phylogeny in Fig. 1A.

## Taxonomy

### Diagnosis of the trichaline clade

Most trichaline genera may be distinguished from other Metriorrhynchini by their general appearance (Figs. 2 and 3) and external characters (Fig. 4). The pronotal carinae are reduced to a single, lanceolate areola in most genera (Figs. 4C–4J and 4M–4T); two divergent pronotal ridges are present in *Eniclases* and five areolae in *Schizotrichalus* (Figs. 4K and 4L). The first primary elytral costa is shortened in all trichaline genera (Figs. 2A, 2B, 2F, 2K, 2Q, 3A, 3D, 3E, 3H and 3L), and in some distantly related Metriorrhynchina, e.g., *Leptotrichalus* and *Kassemia* Bocak, 1998 (*Bocak, 1998a, 2002*). Male genitalia are highly variable, either robust with the characteristic sclerites in the internal sac (*Diatrichalus*; Figs. 5A–5C), the phallus is slender, with a simple sclerotized internal sac (*Lobatang*; Figs. 5D and 5E), robust with the sclerotized base of the internal sac (*Lobatang*; Figs. 5G–5I), slender with the mostly membranous internal sac with a pair of basal thorns (*Trichalus, Microtrichalus*; Figs. 5F and 5J–5L), slender with partly exposed, membranous internal sac (*Flabellotrichalus*; Figs. 5N–5P) or the phallus is

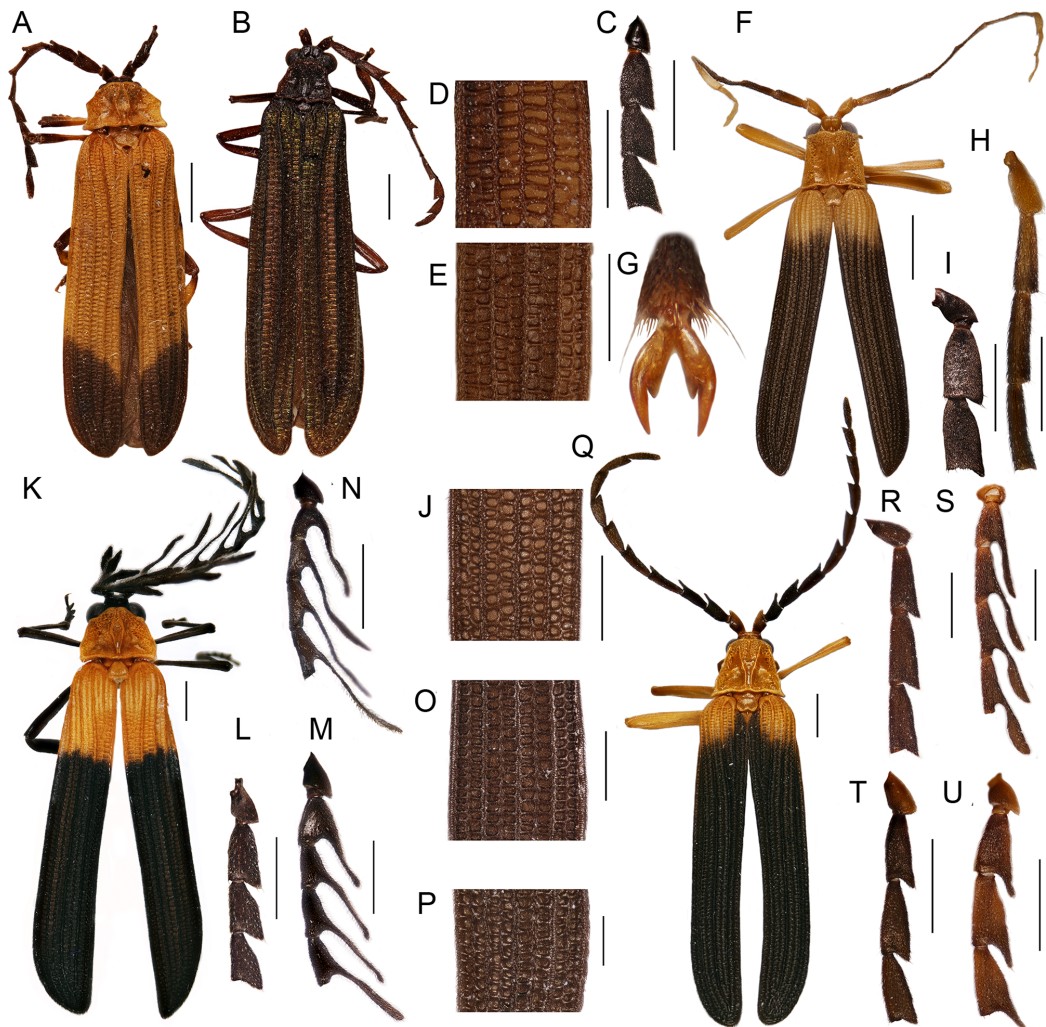

**Figure 2** **General appearance (1).** General appearance, basal male antennomeres, and the posterior part of the right elytron. (A) *Diatrichalus* sp.; (B) *Diatrichalus aeneus* Bocak; (C) *Diatrichalus* sp.; (D) *D. cerberus* (Bourgeois), (E) *D. sinuaticollis* (Pic); (F) *Lobatang* sp.; (G) *L. papuensis* Bocak, hind tarsus claws; (H–J) *Lobatang* spp.; (K) *Flabellotrichalus* sp.; (L) *Flabellotrichalus* sp., female basal antennomeres, (M, N) *Flabellotrichalus* spp., male antennae; (O, P) *Flabellotrichalus* spp.; (Q) *Eniclases divaricatus* Kleine, female; *Eniclases* spp., male antennae: (R) *Eniclases* sp., (S) *E. divaricatus* Kleine, (T) *E. bicolor* Bocek et Bocak, (U) *E. similis* Bocak & Bocakova. Scales 1 mm (A, B, F, K, Q), 0.5 mm (other figures).

almost completely membranous in the apical half and has a characteristic ventral pigmented keel and small cup-shaped apex (*Eniclases*, *Schizotrichalus*; Fig. 5M). The genital morphology of each genus is unique within Metriorrhynchini and enables reliable identification. Female genitalia have dorsally attached vaginal glands in *Schizotrichalus*, *Eniclases*, *Flabellotrichalus*, *Microtrichalus*, and *Trichalus* (Fig. 6Q), but the glands are laterally attached in *Diatrichalus* and *Lobatang* (Figs. 6B and 6E), as in other Metriorrhynchini.

Some trichaline net-winged beetles can be reliably identified only by a combination of characters. The pronotal carinae, elytral ridges and genitalia can be similar in distantly

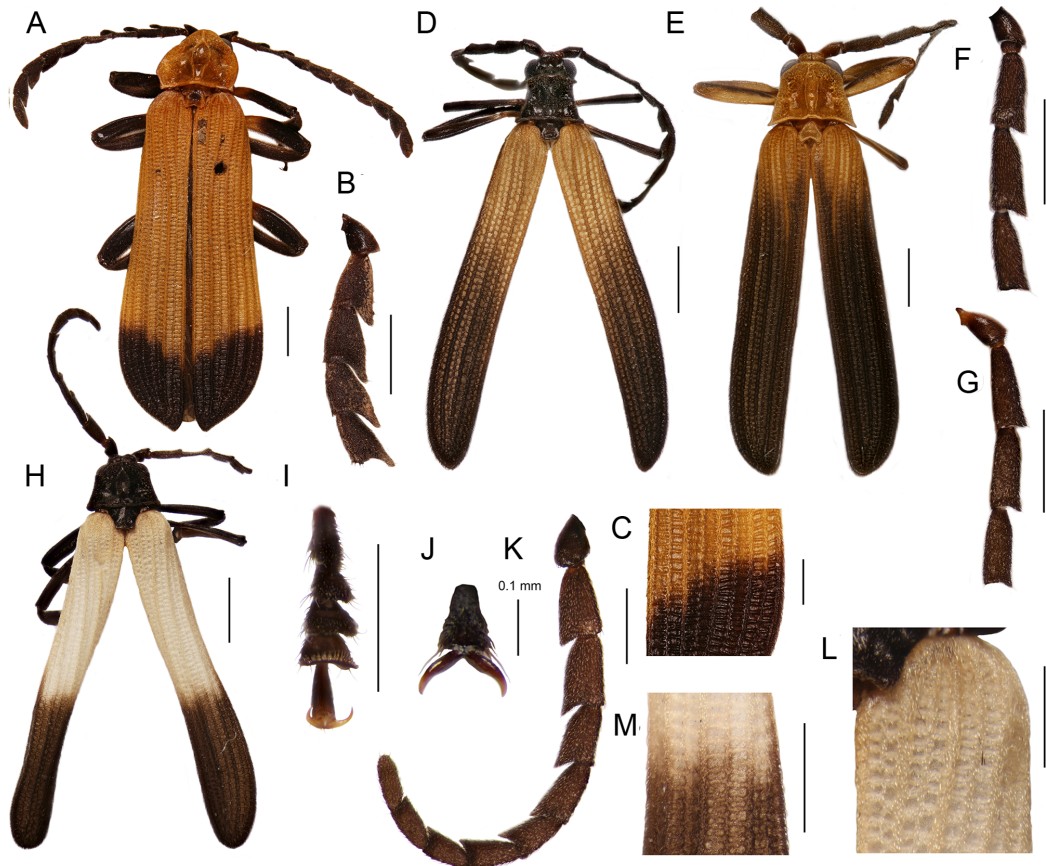

**Figure 3 General appearance (2).** General appearance, basal male antennomeres, and the posterior part of right elytron. (A–C) *Trichalus flavopictus*; (D) *Microtrichalus* sp., male; (E) *Microtrichalus* sp., female; (F, G) *Microtrichalus* spp.; *Flabellotrichalus* (*Maibrius*) *horaki* sp. nov.: (H) general appearance, (I) tarsus, (J) claws, (K) male antenna, (L) humeral part of elytron, (M) middle part of elytron. Scales 1 mm (A, D, E, H), 0.5 mm (B, C, F, G, I, K–M), 0.1 mm (J).

related metriorrhynchine taxa. Therefore, all these structures must simultaneously corroborate the membership in the trichaline clade.

## Redescription

Body small to medium-sized, 4–20 mm long, dorso-ventrally flattened, elytra parallel-sided or slightly widened backwards (e.g., Figs. 2A, 2B and 3A), body mostly dark brown, seldom yellow, upper side variably colored, often with aposematic color patterns combining yellow and dark colored parts; seldom some parts of pronotum and elytra brightly red colored or upper side metallic blue.

Head hypognathous, small, partly hidden by pronotum, rostrum absent in most species, sometimes moderately long rostrum in *Lobatang*. Cranium slightly dorso-ventrally flattened, with more or less prominent antennal tubercles followed by depression; mouth opening approximately as wide as long. Gula wider than long, with more or less wide process, where postmentum is attached; posterior tentorial pits usually unapparent externally; tentorium mostly membranous, only posterior tentorial arm

partly sclerotized. Mandibles relatively stout, short, outer margin covered with dense long setae, sometimes only several short pale setae present. Labrum wider than long, shallowly emarginate apically, with long dense setae. Labium with robust praementum and much smaller u-shaped postmentum. Labial palpi with three palpomeres, palpomere 2 usually longest. Maxillae with long galea; lacinia smaller, sometimes reduced to limited field of pale short setae. Cardo very small, well-sclerotized, movable, stipes flat, with narrow bent inner margin. Maxillary palpi with four palpomeres, palpomeres 1 and 3 always much shorter than palpomeres 2 and 4. Apical palpomeres distally flattened. Antennae with 11 antennomeres, slightly to strongly flattened, antennomere 1 pear-shaped, robust, antennomere 2 very small, antennomeres 3–10 parallel-sided to acutely serrate in both sexes or flabellate in male and serrate in female, antennomere 11 elliptic; antennomeres 3–11 covered with dense, short pubescence.

Pronotum flat, with pronotal carinae (Figs. 4C–4T); *Diatrichalus, Lobatang, Flabellotrichalus, Trichalus,* and *Microtrichalus* with median lanceolate areola, *Eniclases* with two divergent longitudinal carinae (Fig. 4L), and *Schizotrichalus* with three areolae present within the area limited by longitudinal carinae (Fig. 4K). Median areola, if present, either connected with frontal margin by carina or attached directly to frontal and posterior pronotal margins, length of connecting carina variable; sometimes vestigial postero-lateral carinae present close to lateral margins (Figs. 4C–4T). Pronotal surface roughly punctured at frontal and lateral margins; pronotal pubescence usually short, sparse in most species, denser at lateral margins or very long and dense in some *Flabellotrichalus* (Figs. 4O and 4P). Prothoracic pleura concave, with strongly elevated margins, similarly structured as pronotal surface. Prothoracic coxal cavities open. Mesosternum transverse, narrow, bridge-like. Scutellum small, apex shallowly emarginate. Metathorax long, robust, metasternum broad and long, with incomplete midline in distal part.

Elytra flat, parallel-sided to slightly widened backwards, each elytron with nine longitudinal costae at base; four costae robust, called primary costae, intermediate secondary costae weak, sometimes irregular. Primary costa 1 robust only in humeral quarter of elytron, then much weaker, similar to secondary costae; secondary costae between suture and primary costa 1 and between primary costae 1 and 2 missing except humeral quarter of elytron (Figs. 3A, 3D, 3E and 3L); seldom secondary costae absent (some *Diatrichalus*; Fig. 2D).

Abdomen flat, free, with eight visible sternites in male and seven in female. Shape of male terminal sternites variable, affected by shape of phallus. Subapical male abdominal sternite more or less emarginate at hind margin. Last visible tergite long, spoon-like, often with small sclerotized tergite attached to inner surface, this tergite sometimes membranous, undetectable. Female terminal abdominal segments variable in shape and most species with short spiculum gastrale (Figs. 6C, 6D, 6I, 6J, 6L, 6M and 6R).

Male genitalia variable in shape (Figs. 5A–5P). Phallobase circular, subtle, with more or less extensive membrane, membrane soft to lightly sclerotized. Parameres absent, phallus mostly slender, with well-sclerotized or partly membranous apical part, open ventrally

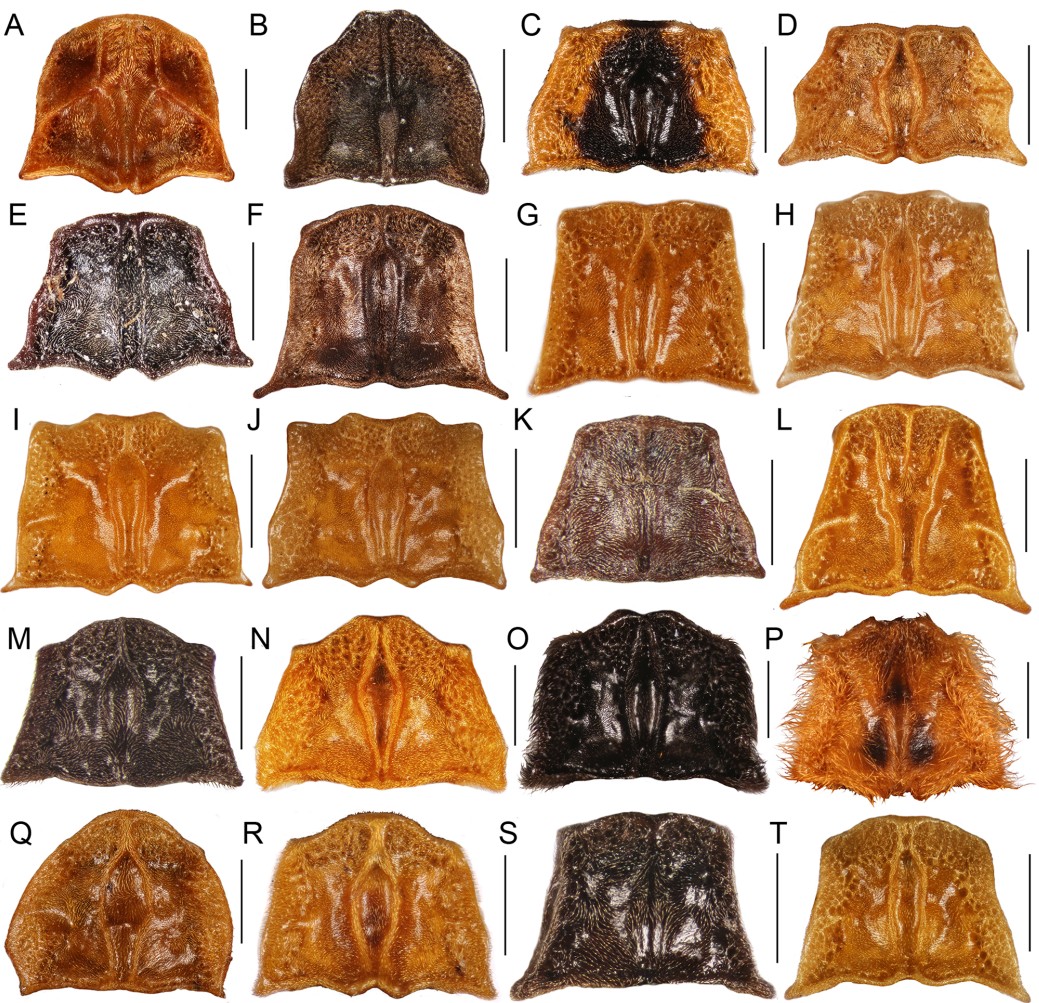

**Figure 4 Pronota.** (A) *Metriorrhynchus inaequalis* (F.); (B) *Bulenides* sp.; (C) *Diatrichalus* sp.; (D) *D. mancus* (Kleine); (E) *D. aeneus* Bocak; (F) *Lobatang papuensis* Bocak; (G–J) *Lobatang* spp.; (K) *Schizotrichalus* sp.; (L) *Eniclases divaricatus* Kleine; (M–P) *Flabellotrichalus* spp.; (Q) *Trichalus flavopictus* Waterhouse; (R) *T. communis* Waterhouse; (S, T) *Microtrichalus* spp. Scales 0.5 mm.

with exposed internal sac. Internal sac membranous to sclerotized, with apical complex sclerite or with pair of slender sickle-like thorns at base.

Ovipositor mostly with long, slender valvifers (Figs. 6A, 6H, 6K, 6O and 6P), sometimes valvifers connected at their bases by membrane, which can be sclerotized in high degree; seldom valvifers basally fused with coxites (Fig. 6E). Valvifers robust, connected in basal third in some *Trichalus*. Vagina slender, paired glands inserted laterally (Fig. 6B) or dorsally (Fig. 6Q). Bases of glandular ducts slender, seldom robust (*Trichalus*), but regularly more sclerotized than terminal gland, flat unpaired gland in terminal part of vagina, lateral pockets and slender unpaired basal gland in *Microtrichalus* (Fig. 6H). Spermatheca long, and slender (Fig. 6B), lemon-shaped, with spirally coiled spermaduct; y-shaped gland attached to apex of spermatheca (Fig. 6K).

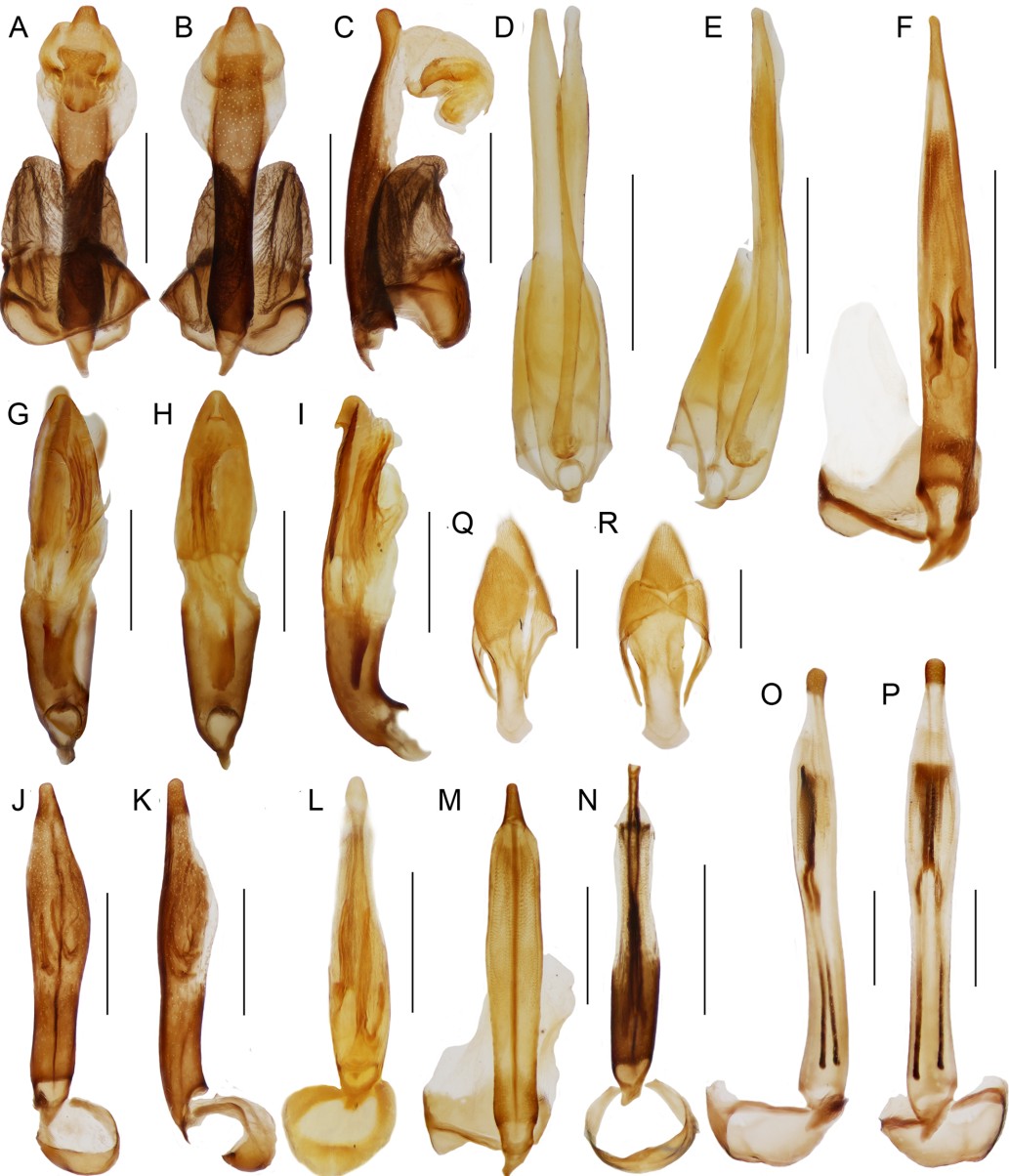

**Figure 5 Male genitalia.** Male genitalia and terminal abdominal sclerites. (A–C) *Diatrichalus* sp.; (D, E) *Lobatang* sp.; (F) *Trichalus flavopictus* Waterhouse; (G–I) *Lobatang* sp.; (J, K) *Trichalus* sp.; (L) *Micro-trichalus* sp.; (M) *Eniclases* sp.; (N) *Flabellotrichalus* (*Maibrius*) *horaki* sp. nov.; (O, P) *Flabellotrichalus* sp.; (Q, R) *Lobatang* sp., male terminal abdominal sclerites, ventrally and dorsally. Scales 0.5 mm.

### *Diatrichalus Kleine, 1926*

(Figs. 2A–2E, 4C–4E, 5A–5C and 6A–6D)

*Diatrichalus Kleine, 1926*: 167.

**Type species:** *Diatrichalus xylobanoides Kleine, 1926*, by original designation.
=*Mimotrichalus Pic, 1930*: 92, hors texte; *Bocak, 1998a*: 182.

**Type species:** *Mimotrichalus tenimberensis Pic, 1930*, by monotypy.

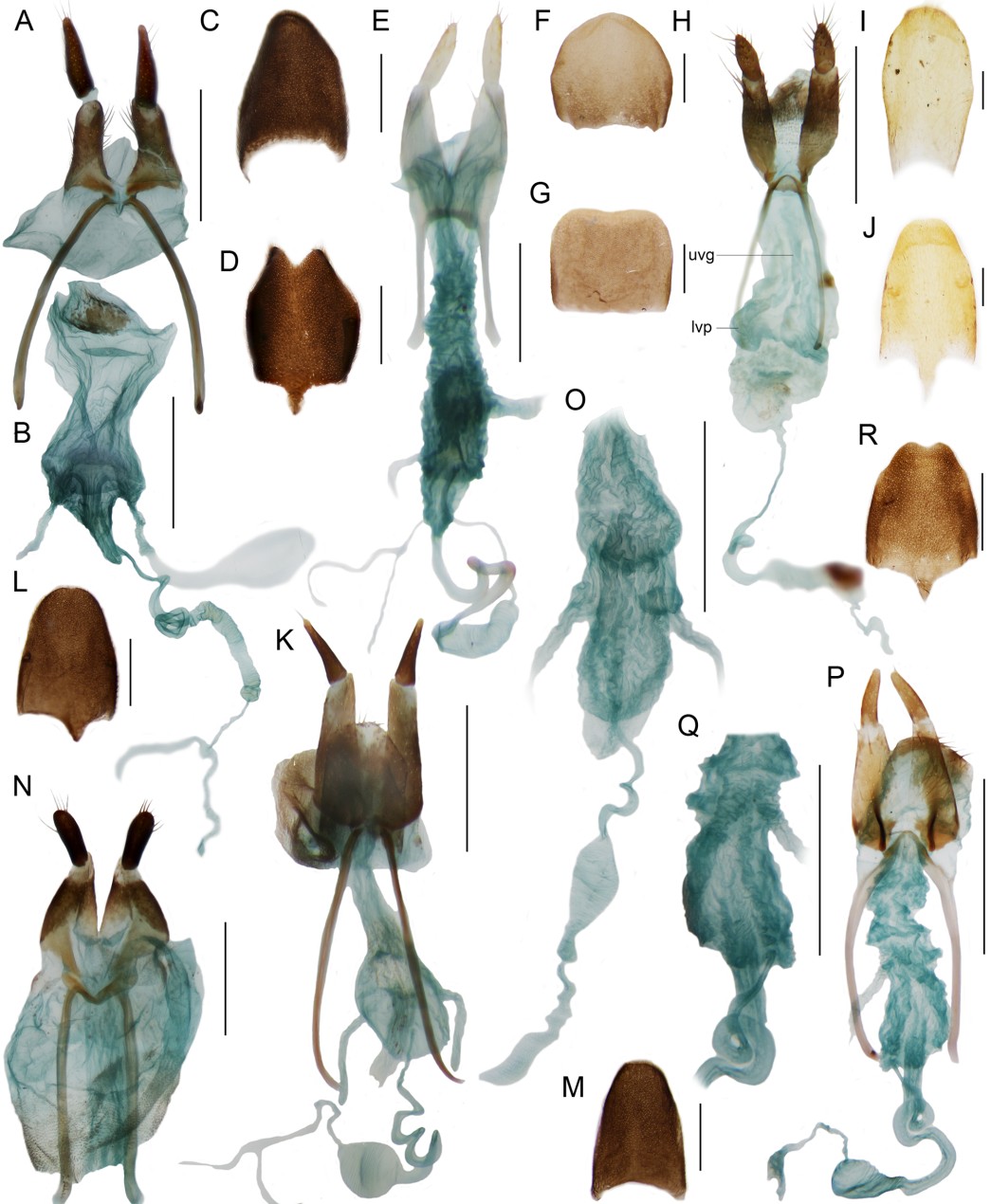

**Figure 6 Female genitalia.** Female genitalia and terminal abdominal sclerites. *Diatrichalus* sp. (A) ovipositor, (B) female genitalia, (C) terminal tergite, (D) terminal sternite; *Lobatang* sp. (E) ovipositor and female genitalia, (F) terminal sternite, (G) terminal tergite; *Microtrichalus* sp. (H) ovipositor and female genitalia, (I) terminal tergite, (J) terminal sternite; *Flabellotrichalus* sp. (K) ovipositor and female genitalia, (L) terminal sternite, (M) terminal tergite; *Trichalus* sp. (N) ovipositor, (O) female genitalia; *Eniclases divaricatus* Kleine (P) ovipositor, (Q) vagina, dorsally, (R) terminal sternite; uvg, unpaired gland; lvp, lateral vaginal pocket. Scales 0.5 mm.

**Diagnosis:** Pronotum with median, often wide areola, lateral carinae absent or very obtuse (Figs. 4C–4E), antennae of both sexes more or less acutely serrate to shortly flabellate (Fig. 2C), phallus stout, apical part projected, internal sac more or less

sclerotized (Figs. 5A–5C), vaginal glands inserted laterally, valvifers free, slender, spermatheca long, slim (Figs. 6A and 6B), tarsal claws simple.

**Remark:** *Kleine (1926)* restricted *Diatrichalus* to species with four elytral costae, as in *D. xylobanoides* (Fig. 2D), and Pic described *Mimotrichalus* as having additionally obtuse, irregular and commonly interrupted secondary costae. The current concept of *Diatrichalus* is wide and includes all species with four and nine costae and their intermediate forms (Figs. 1A, 2D and 2E; *Bocak, 2001*). Our molecular dataset contained only a single species without secondary elytral costae, *D. xylobanoides*, which is a sister species to other *Diatrichalus*, included in the analyses. The current results support two clades which correspond with earlier concepts of *Diatrichalus* and *Mimotrichalus*, but *Bocak (2001)* showed that other species without secondary costae have diverse genitalia, and we suppose that if these are included in future phylogenetic analyses they will not form a monophylum. Additionally, there are multiple species with gradual reduction of secondary costae and they can only be arbitrarily assigned to their respective groups. Therefore, we propose to keep *Mimotrichalus* in the synonymy of *Diatrichalus*. Although the antennae have never long lamellae, they are sometimes so acutely serrate that *Kleine (1933b)* classified *D. salomonensis* (*Kleine, 1933b*) in *Flabellotrichalus* (*Bocak, 2001*).

***Lobatang* *Bocak, 1998a***

(Figs. 2F–2I, 4F–4J, 5D, 5E, 5G–5I and 6E–6G)

*Lobatang* *Bocak, 1998a*: 190.

**Type species:** *Lobatang papuensis* *Bocak, 1998a*.

**Diagnosis:** Antennomeres 3–10 parallel-sided to serrate (Figs. 2H and 2I), pronotum with median lanceolate areola (Figs. 4F–4J), male genitalia variable in shape, always with sclerotized base of internal sac (Figs. 5G–5I) or whole internal sac sclerotized and long (Figs. 5D and 5E), tarsal claws split (Fig. 2G).

**Remark:** The clade *Leptotrichalus* + *Lobatang* was based on the shape of valvifers (*Bocak, 1998a*, *2002*), but the molecular phylogeny indicates the distant position of these genera (Fig. 1A; *Sklenarova, Kubecek & Bocak, 2014*).

***Lobatang* s. str.**

**Type species:** *Lobatang papuensis* *Bocak, 1998a*.

**Diagnosis:** The nominotypical subgenus differs from *Spinotrichalus* only in the absence of femoral and tibial thorns in hind legs.

**Subgenus *Spinotrichalus* *Kazantsev, 2010*, stat. nov.**

*Spinotrichalus* *Kazantsev, 2010*: 93.

**Type species:** *Spinotrichalus telnovi* *Kazantsev, 2010*, by original designation.

**Diagnosis:** As the nominotypical subgenus, but hind femora and tibiae with small thorns.

**Remark:** *Kazantsev (2010)* described *Spinotrichalus*, which shares very similarly shaped genitalia and split claws with *Lobatang*. Besides the body shape and coloration, the type

species of *Spinotrichalus* and *Lobatang* differ only in the presence of femoral and tibial thorns. This character is the autapomorphy of *S. telnovi* and *Spinotrichalus* may be treated as a synonym, if its position renders *Lobatang* paraphyletic. As the type species of both genera are unavailable for DNA analysis, we prefer to keep *Spinotrichalus* as a valid name till more data are available. Based on highly similar male genitalia (Figs. 5D and 5E; *Kazantsev, 2010*), we lower its rank to a subgenus of *Lobatang Bocak, 1998a*. Consequently, the new combination *Lobatang (Spinotrichalus) telnovi (Kazantsev, 2010)* is proposed.

### *Eniclases Waterhouse, 1879*

(Figs. 2Q–2U, 4L, 5M, 6P and 6R)

*Eniclases Waterhouse, 1879*: 66.

**Type species:** *Lycus* (genus 35) *luteolus Waterhouse, 1878*, by original designation.
  =*Trichalolus Pic, 1923*: 36, hors texte; *Bocak & Bocakova, 1991*: 206.

**Type species:** *T. apertus Pic, 1923*, by monotypy.

**Diagnosis:** Pronotum with two longitudinal divergent carinae dividing pronotum in three fields (Fig. 4L), phallus very slender with pigmented dorsal keel, internal sac without thorns; whole internal sac membranous (Fig. 5M); lateral vaginal glands dorsally attached (as in Fig. 6Q).

**Remark:** The *Eniclases* male antennae are highly variable in shape and several species have acutely serrate to flabellate antennae (Figs. 2R–2U; *Bocak & Bocakova, 1991*; *Bocek & Bocak, 2016*). Only one of these species was included in the molecular analysis and it was recovered as a sister to its congeners (Fig. 1A). Other morphological characters and molecular phylogeny indicate that the species with similar antennae are not closely related (*Bocek & Bocak, 2016*; *Bocak & Bocakova, 1991*). Therefore, we do not consider this character to be valuable in the delimitation of a genus or subgenus in this clade.

### *Schizotrichalus Kleine, 1926*

(Fig. 4K)

*Schizotrichalus Kleine, 1926*: 167.

**Type species:** *T. nigrescens Waterhouse, 1879*, by original designation.

**Diagnosis:** Pronotum with five areolae (Fig. 4K), phallus with pigmented dorsal keel, internal sac without thorns; vaginal lateral glands dorsally attached.

**Remark:** *Schizotrichalus* was unavailable for molecular analyses and was inferred as a genus closely related to *Eniclases* in the morphology-based phylogeny (Figs. 1B and 1C; *Bocak, 1998a*, *2002*).

### *Flabellotrichalus Pic, 1921b*

(Figs. 2K–2P, 3H–3M, 4M–4P, 5N–5P and 6K–6M)

*Flabellotrichalus Pic, 1921b*: 9, hors texte.

**Type species:** *Flabellotrichalus notatithorax* Pic, 1921, subsequent designation, *Kleine (1936)*.
  =*Stereotrichalus Kleine, 1926*: 183; *Kleine, 1930*: 330.

**Type species:** *Stereotrichalus evidens Kleine, 1926*, by monotypy.
  =*Villosotrichalus Pic, 1921b*: 9, hors texte; *Bocak, 1998a*: 183.

**Type species:** *Villosotrichalus reductus Pic, 1921b*, by monotypy.

**Diagnosis:** Male antennae flabellate (Figs. 2M and 2N) or seldom serrate (Fig. 3K), pronotum with single longitudinal median areola, frontal and lateral margins of pronotum often with dense short to very long pubescence (Figs. 4M–4P), phallus very slender, internal sac without thorns; whole internal sac membranous with y-shaped base (Figs. 5N–5P); lateral vaginal glands attached dorsally.

**Remark:** The molecular phylogeny recovered a species with dense pronotal pubescence in the terminal position (Fig. 1A) which supports the earlier synonymization of *Villosotrichalus* to *Flabellotrichalus* (*Bocak, 1998a*).

**Subgenus *Flabellotrichalus Pic, 1921b***

**Diagnosis:** All diagnostic characters as in the whole genus, but the male antennae are always flabellate (Figs. 2M and 2N).

**Classification and distribution:** *Flabellotrichalus* occur in Australia, New Guinea, and the Moluccas. Nine Australian and New Guinean species were included in current analyses, but none was identified to the species level due to chaotic alpha-taxonomy (Fig. 1). The genus has never been revised and all 15 formally described species are known only from original descriptions. Two species with dense pronotal pubescence were classified originally as *Villosotrichalus* and this genus was synonymized with *Flabellotrichalus* (Bocak, 1998). The species similar to the typical *Villosotrichalus* were inferred in the terminal position within *Flabellotrichalus* in current analyses (Fig. 1A).

**Subgenus *Maibrius* subgen. nov.**
LSID: urn:lsid:zoobank.org:act:0A2E45FB-72DB-49E7-BD7C-BC792072B106
(Figs. 3H–3M, 4M and 5N)

**Type species:** *Flabellotrichalus (Maibrius) horaki* sp. nov.

**Diagnosis:** Male antennae serrate (Fig. 3K), pronotum with single longitudinal median areola, frontal and lateral margins of pronotum with dense short pubescence (Fig. 4M), phallus slender, apically membranous; internal sac without thorns, membranous, with y-shaped base (Fig. 5N); lateral vaginal glands attached dorsally. *Maibrius* subgen. nov. differs from the nominotypical subgenus in the serrate male antennae (Fig. 3K) and shorter, relatively robust phallus (Fig. 5N).

**Remark:** The molecular phylogeny identified *F. (Maibrius) horaki* sp. nov. as a genetically distant sister-lineage to other *Flabellotrichalus* (Fig. 1A). This species cannot be identified as a close relative of *Flabellotrichalus* without dissection of male genitalia or DNA sequencing. The general appearance and morphology of antennae resemble *Trichalus* or *Microtrichalus* and only the male genitalia indicate relationships to *Flabellotrichalus*. This conservative taxonomy keeps *Flabellotrichalus* s. str. morphologically well-defined and reflects the genetic and phenotypic divergence of *F. (Maibrius) horaki* sp. nov. Female remains unknown.

**Etymology:** The subgeneric name is derived from the name "Maibri," a village in the Arfak mountains where the type species was collected. The genus name is the noun of masculine gender.

*Flabellotrichalus* (*Maibrius*) *horaki* sp. nov.
LSID: urn:lsid:zoobank.org:act:86069ACA-BC85-4865-847B-2EB421DC3BC3
(Figs. 3H–3M, 4M and 5N)

**Type material:** Holotype. Male, "New Guinea, West Papua prov., Arfak Mts., Maibri vill., 2015, local coll." (GenBank Voucher Number UPOL BM0082; deposited in the collection of the Palacky University in Olomouc, Czech Republic, LMBC).

**Diagnosis:** *Flabellotrichalus* (*Maibrius*) *horaki* sp. nov. differs from all known *Flabellotrichalus* in the serrate male antennae (Fig. 3K). Its phallus is slightly more robust than in other *Flabellotrichalus* (Figs. 5N–5P). *F.* (*M.*) *horaki* sp. nov. is currently a single trichaline species with white colored humeri.

**Description:** Male. Body 7.8 mm long, dorso-ventrally flattened, relatively slender, dark brown to black, only basal three fifths of elytra pale yellow to white colored (Fig. 3H). Head small, eyes small-sized, hemispherically prominent, eye diameter 0.64 times interocular distance; antennae serrate (Fig. 3K). Pronotum 1.24 wider than long at midline, trapezoidal, widest at base, anterior angles almost rectangular, well-marked, lateral margins slightly concave, posterior angles sharply prominent; areola wide, connected with anterior margin by short carina, lateral carinae completely absent, disc of pronotum roughly sculptured at frontal and lateral margins, covered with dense, short pubescence (Fig. 4M). Elytra with three primary and four secondary costae in middle part of elytron, elytra 3.7 times longer than width at humeri, rectangular cells dense, irregular, costae covered with dense pubescence (Figs. 3L and 3M). Phallus relatively short, sclerotized and pigmented in basal two fifths, apical part membranous, with a cup-shaped apex held by pair of pigmented keels; internal sac membranous, with y-shaped, pigmented base, without any thorns (Fig. 5N). Legs flattened, densely pubescent, tarsi wide (Fig. 3I), claws simple (Fig. 3J). Female unknown.

**Measurements:** Body length 7.8 mm, pronotum length 0.91 mm, pronotum width 1.13 mm, width at humeri 1.75 mm, length of elytron 6.55 mm, eye diameter 0.38 mm, eye distance 0.59 mm, length of phallus 1.14 mm.

**Etymology:** The specific name is a patronym in honor of Jan Horak, a Czech specialist in Mordellidae.

**Distribution:** New Guinea, Arfak mountains.

*Trichalus* Waterhouse, 1877
(Figs. 3A–3C, 4Q, 4R, 5F, 6N and 6O)
*Trichalus* Waterhouse, 1877: 82.

**Type species:** *T. flavopictus* Waterhouse, 1877, subsequent designation, Waterhouse, 1878: 103.
=*Xantheros* Fairmaire, 1877: 167; Bourgeois, 1891: 347.

**Type species:** *Xantheros ochreatus Fairmaire, 1877*.

**Diagnosis:** Antennae serrate in both sexes, pronotum with single longitudinal median areola, apical part of phallus commonly well-sclerotized (Figs. 5F, 5J and 5K), internal sac with two thorns; lateral vaginal glands attached dorsally, valvifers free or connected basally (Fig. 6N) or sub-basally, forming H-shaped structure in some species, tarsal claws simple, vaginal lateral pockets and unpaired basal gland absent.

**Remark:** The type *of X. ochreatus*, the type species of *Xantheros*, was very probably destroyed (Bocak, 1998a). The original publication cites "Sydney" as the type locality and although we had at our disposal the extensive collection of Australian trichaline net-winged beetles from ANIC (Canberra), we found no specimen whose morphology agrees to the original description and originates from southern New South Wales. Similar species occur only in northern New South Wales and in Queensland. As we are not able to designate the neotype, we keep *Xantheros* in synonymy of *Trichalus* (Kleine, 1933a; Bocak, 1998a, 2002).

### *Microtrichalus Pic, 1921b*
(Figs. 3D–3G, 4S, 4T, 5L and 6H–6J)
*Microtrichalus Pic, 1921b*: 9 (hors texte).

**Type species:** *M. singularis Pic, 1921b*, by monotypy.
    =*Falsoenylus Pic, 1926*: 29, hors texte; Bocak, 1998a: 184.

**Type species:** *F. basipennis Pic, 1926*, by monotypy.

**Diagnosis:** Antennae weakly serrate in both sexes, pronotum with single longitudinal median areola, apical part of phallus weakly sclerotized, internal sac with two thorns, lateral vaginal glands attached dorsally, vagina with two lateral pockets situated in middle of vaginal length and very slim, long, unpaired gland between valvifers (Fig. 6H), valvifers slender, sometimes fused basally.

### Key to the genera and subgenera of the trichaline clade

1. Tarsal claws split (Fig. 2G), *Lobatang Bocak, 1998a* ..................................... 2
   —Tarsal claws simple (Fig. 3J) ..................................................... 3

2. Male hind femora and tibiae without any thorn ......... **Lobatang** (**Lobatang** s. str.)
   —Male hind femora and tibiae with small thorns.......... **Lobatang** (**Spinotrichalus** *Kazantsev, 2010*)

3. Apical margins of maxillary and labial palpomeres with sensillae, apical palpomeres securiform, apical part of phallus robust, internal sac complex, partly sclerotized (Figs. 5A–5C); vaginal glands inserted laterally (Fig. 6B), basal part of spermaduct wide, spermatheca long, slender ............................... **Diatrichalus** *Kleine, 1926*
   —Apical margins of maxillary and labial palpomeres without sensillae, apical palpomeres variable shaped, apical part of phallus slender, internal sac membranous or with a pair of sickle shaped thorns (Figs. 5F and 5J–5P); vaginal glands inserted dorsally (Fig. 6Q), basal part of spermaduct slender, spermatheca bulbous (Figs. 6H, 6K, 6O and 6P) .......................................................................... 4

4. Pronotum with five areolae or with two anteriorly divergent longitudinal carinae (Figs. 4K and 4L), phallus with single pigmented dorsal keel (Fig. 5M) ............ 5
   —Pronotum with single lanceolate longitudinal areola attached to frontal and basal margin of pronotum at a single point (Figs. 4M–4T), pigmented dorsal keel absent in most species (Figs. 5F and 5N–5P, but compare with Fig. 5J) ...................... 6

5. Pronotum with five areolae (Fig. 4K) ................... **Schizotrichalus** *Kleine, 1926*
   —Pronotum with two divergent longitudinal carinae (Fig. 4L)............. **Eniclases** *Waterhouse, 1879*

6. Male antennae flabellate ................. **Flabellotrichalus** (**Flabellotrichalus** s. str.)
   —Male antennae serrate or antennomeres parallel-sided .......................... 7

7. Internal sac membranous, without thorns, with pigmented y-shaped basal part (*Maibrius* females are unknown) ........ **Flabellotrichalus** (**Maibrius** subgen. nov.)
   —Internals sac with two thorns ............................................... 8

8. Vagina with two lateral pockets in middle part and with unpaired slim and long basal gland (Fig. 6H), valvifers slender, usually free, sometimes connected basally .............................................. **Microtrichalus** *Pic, 1921b*
   —Vagina without lateral pockets and unpaired gland, valvifers often robust, connected basally or sub-basally (Fig. 6N) ......................... **Trichalus** *Waterhouse, 1877*

## DISCUSSION

We present the first densely sampled molecular phylogeny and separate morphological analyses of all genera which were traditionally placed in the trichaline clade (Figs. 1A–1C). The terminal position of the trichaline clade in Metriorrhynchina has already been demonstrated in the molecular analyses of Metriorrhynchini, and trichaline genera lost their formal rank in classification (*Sklenarova, Kubecek & Bocak, 2014*). Our analyses of the current more extensive dataset confirm the terminal placement of the trichaline clade within Metriorrhynchina (Fig. 1A). Metriorrhynchina are well-supported as a monophylum in all previous analyses (*Bocak et al., 2008*; *Sklenarova, Chesters & Bocak, 2013*; *Sklenarova, Kubecek & Bocak, 2014*), therefore, Cautirina were used as an outgroups and Metriorrhynchina, here consisting of trichaline terminals and 17 non-trichaline terminals, were forced by a single outgroup to be monophyletic. Such dataset is fully capable to test if trichaline genera are a sister lineage of other Metriorrhynchina or a terminal lineage within this subtribe as in all earlier analyses (*Bocak et al., 2008*; *Sklenarova, Chesters & Bocak, 2013*; *Sklenarova, Kubecek & Bocak, 2014*).

The (*Leptotrichalus*, (*Synchonnus, Wakarumbia*)) clade is a sister lineage to trichaline genera in the molecular analyses although with ambiguous support (BS 23%; PP 0.98; Fig. 1A). *Leptotrichalus* and *Synchonnus* were earlier placed in the trichaline clade, but *Wakarumbia* differs substantially in the presence of unique five-areolae in the pronotum, full-length elytral costae, and the morphology of genitalia (*Bocak, 2002*). Therefore, an expansion of the trichaline clade would be impractical.

Four trichaline genera are included in our molecular analyses for the first time and now six of seven genera are represented in the DNA data set: *Diatrichalus* and *Lobatang* are

members of the trichaline clade as defined here and they are deeply rooted lineages in close relationships to the earlier narrowly defined trichaline clade (*Bocak, 1998a*, *2002*). *Eniclases* is a sister to the clade ((*Flabellotrichalus*, *Trichalus*), *Microtrichalus*) (Fig. 1A).

The morphological analyses indicate different relationships. They suggest a topology which contains the clades (*Synchonnus + Diatrichalus*) and (*Leptotrichalus + Lobatang*) in contrast with molecular analyses (Fig. 1A; *Sklenarova, Kubecek & Bocak, 2014*). Such relationships are supported by the similar shape of pronotal carinae in trichaline genera, *Synchonnus*, and *Leptotrichalus* and the shortened elytral costa 1 in all genera except *Synchonnus*. Due to the limited number of other informative phenotypic characters, the homology of these character states cannot be falsified in the current morphological analyses (Figs. 1A–1C). The single lanceolate areola and the shortened elytral costa 1 were present in the most recent common ancestor of the trichaline clade (Fig. 1A), but similar arrangements of pronotal carinae and elytral costae have been found in several unrelated taxa, e.g., the shortened costa in *Kassemia* and the similar pronotum in some *Cautires* (*Bocak, 2002*; *Sklenarova, Kubecek & Bocak, 2014*). The high plasticity of pronotal carinae is additionally indicated by a hypothesized reversal in *Eniclases* and *Schizotrichalus* (Fig. 1A). Therefore, we consider the phylogenetic signal provided by these external characters to be unreliable and male and female genitalia should be studied to verify recovered relationships.

The molecular topology regularly indicates a deep position of *Diatrichalus* and *Lobatang*, but we have not been able to find any phenotypic character which supports their relationships with other trichaline genera, except for the above mentioned lanceolate pronotal areola and the shortened elytral costa 1. Conversely, the monophyly of the restricted trichaline clade, i.e., *Eniclases + Flabellotrichalus + Trichalus + Microtrichalus* is supported by unique, dorsally attached vaginal glands (Fig. 6Q) in the morphological analysis, but their relationships, although simultaneously recovered by molecular analyses, had only a low statistical support (BS 74%, PP 0.48). The internal relationships within this clade were better resolved in the DNA-based topology, which indicates the deeply rooted position for *Eniclases* with respect to other genera of the restricted trichaline clade (Figs. 1A–1C). *Schizotrichalus* was not available for the molecular analyses and its close relationships with *Eniclases* are based on morphology (Figs. 1B and 1C). *Trichalus* and *Flabellotrichalus* form a clade with a low support in molecular analyses (BS 64%, 0.92 PP) and their sister position has never been inferred from morphology (Figs. 1A–1C). Their relationship is supported by similar pigmented keels at the apex of the phallus in some species, but no other character (Figs. 5F and 5N–5P). In contrast, *Microtrichalus* and *Trichalus* share sickle-shaped thorns in the basal part of their internal sac (Figs. 5F and 5J–5L). Concerning the low bootstrap support, these relationships need further data to be validated. Additionally, *Trichalus* is not assuredly monophyletic (Fig. 1A) and may split into several clades if more taxa are included in future analyses. The absence of a synapomorphy which supports *Trichalus* also complicates identification. Some species cannot be reliably identified as *Trichalus* without information on female genitalia. *Microtrichalus* has unique pockets in the middle part of the vagina and an unpaired basal vaginal gland (Fig. 6H). Both structures are absent in *Trichalus*.

For a long time, the phenotypic diagnoses of most trichaline genera were ambiguous. *Trichalus* served as a basket where most species were placed, and numerous species were later transferred to *Diatrichalus*, *Lobatang*, and *Microtrichalus* (*Kleine, 1926*; *Bocak, 1998a*, *2000*, *2001*). Now, the generic limits are much better defined than in the original descriptions and concepts applied by M. Pic and R. Kleine (*Kleine, 1926*; *Pic, 1921b*, *1923*, *1926*, *1930*), but even with these revised morphological diagnoses, the evaluation of external phenotypic characters is generally insufficient and dissection of genitalia is needed for reliable generic placement.

Some phenotypic characters are affected by the natural and sexual selection and they can rapidly evolve (*Bocek & Bocak, 2016*; *Frazee & Masly, 2015*). Hence, they may provide a misleading phylogenetic signal. Below, we discuss some characters with regard to their diagnostic value and congruence with molecular phylogeny.

**The shape of male antennae**
Filiform, serrate and flabellate male antennae have been used as diagnostic characters, but their value is questioned by variable morphology in related species (e.g., *Cautires*; *Sklenarova, Kubecek & Bocak, 2014*). A high variability in the shape of male antennae was observed in *Lobatang* (Figs. 2H and 2I) and *Eniclases* (Figs. 2R–2U); other genera, such as *Microtrichalus*, have quite uniform antennae (Figs. 3F and 3G). The present study supports the earlier finding that the serrate and flabellate antennae can evolve repeatedly. *Diatrichalus salomonensis* (*Kleine, 1933b*) and some species of *Eniclases* (Figs. 1A and 2R–2U) have very acutely serrate to flabellate antennae, unlike the congeneric species. *Flabellotrichalus* s. str. is well-delimited by the flabellate antennae. We identified a single species, *F.* (*Maibrius*) *horaki* sp. nov., which differs in the serrate male antennae and is also genetically distant from other *Flabellotrichalus*. It was recovered as a sister to the extensive clade of *Flabellotrichalus* s. str. The antennae are an olfactory organ and selection for a large surface can be responsible for rapid morphological evolution in some terminal lineages.

**The shape of the pronotum and pronotal carinae**
The shape of the pronotum is commonly used for morphological identification of net-winged beetle genera and some trichaline species can be assigned to a genus using pronotal morphology. The densely pubescent pronotal margins are characteristic for some but not all *Flabellotrichalus* (Figs. 4O and 4P). Transverse pronota with a large median areola and uniquely shaped lateral margins are characteristic for some *Diatrichalus* (Fig. 4D), but these traits are inconspicuous in some congeneric species (Figs. 4C and 4E). Similarly, the flat pronotum with the characteristic shape of the frontal margin and almost rectangular anterior angles is typical of some, but not all, *Lobatang* (Figs. 4F–4J). The shape of the pronotum is affected by the general appearance (e.g., Figs. 3D and 3E). Net-winged beetles are often associated with mimicry rings and substantially different body sizes, shapes and colorations were identified in recently split sister species, e.g., in *Eniclases* and *Synchonnus* (*Bocek & Bocak, 2016*; *Kusy, Sklenarova & Bocak, in press*). Therefore, these characters, although sometimes useful for quick identification, are

generally unreliable, as can be demonstrated by similar pronota in several species of *Lobatang* (Fig. 4F), *Flabellotrichalus* (Fig. 4M), *Trichalus* (Fig. 4R), and *Microtrichalus* (Figs. 4S and 4T).

An earlier study has already demonstrated that the unique arrangement of seven pronotal areoles is an ancestral state in Metriorrhynchina (Fig. 4A; *Sklenarova, Kubecek & Bocak, 2014*). Although numerous species have the full number of seven areoles (Fig. 4A; *Cautires*, *Metriorrhynchus Gemminger & Harold, 1869*, *Porrostoma Castelnau, 1838*, and others) or their reduction is so limited that the original pattern can easily be recognized (some *Cautires*; *Jiruskova, Motyka & Bocak, 2016*), there are numerous genera with considerably simplified pronotal carinae. When these reduced patterns are considered to be homologous, they lead to a false phylogenetic placement and classification, as occurred when the monophyly was hypothesized and the genus-rank given to *Bulenides*, now placed in *Cautires* (Fig. 4B; *Dudkova & Bocak, 2010*) and also when an independent position and high rank were proposed for trichaline genera (*Kleine, 1928*, *1933a*; *Bocak, 1998a*, *2002*). The earlier defined family rank taxon for trichaline genera, including *Leptotrichalus* (*Kleine, 1928*, *1933a*), was defined by a single areola in most genera: the wide areola in *Diatrichalus* (Figs. 4C and 4D), the very slender areola in *Leptotrichalus*, and a single narrow areola in *Microtrichalus* and *Trichalus* (Figs. 4Q–4T). A similar single areola has been identified in distantly related net-winged beetles, such as Afrotropical Slipinskiini, which had been considered congeneric with the Australian metriorrhynchine genus *Stadenus Waterhouse, 1879* (*Kleine, 1933a*). Similarly, the arrangement of pronotal carinae in some *Synchonnus*, a genus related to *Falsolucidota Pic, 1921a* and *Wakarumbia*, provided a misleading signal for the placement of an earlier valid *Enylus* into close relationships with the trichaline genera (Figs. 1B and 1C; *Bocak, 2002*; *Kusy, Sklenarova & Bocak, in press*). The complex structures are considered to be better indicators of relationships, but in the case of *Eniclases* and *Schizotrichalus*, unique characteristic pronotal patterns, apparently resembling the complex ancestral arrangement (Figs. 4K and 4L), were recovered in the terminal lineage of the trichaline clade in which all close relatives lost the fronto-lateral pronotal carinae (Figs. 1A–1C and 4A–4T). Our results suggest that variable arrangements of pronotal carinae can evolve through reductions in unrelated lineages and, surprisingly, also through the re-appearance of earlier lost structures. These facts indicate the low explanatory power of this character for phylogenetic inference and generic classification (Fig. 1A).

## Elytral costae

Elytral costae were traditionally considered to be reliable characters for generic phenotypic diagnoses in net-winged beetles (*Pic, 1923*, *1930*; *Kleine, 1926*). The concept of *Diatrichalus* was originally based on the presence of four longitudinal elytral costae, in contrast with nine costae in other trichaline genera (*Kleine, 1926*; *Pic, 1930*). The generic limits of this genus were redefined using genitalia, and the loss of secondary costae is assumed in several unrelated species (*Bocak, 2001*). The present DNA dataset contains only a single *Diatrichalus* with absent secondary costae (Fig. 1A). A similar loss of secondary costae was identified in some Afrotropical *Cautires*

(*Sklenarova, Kubecek & Bocak, 2014*) and in an undescribed species of *Schizotrichalus*. Net-winged beetles are soft-bodied and therefore the elytral costae apparently have a strengthening function. The arrangement of the costae depends on body size and shape. The costae are commonly reduced in species with very slender or small bodies such as in Dilophotes (Lycidae: Dilophotini; *Bocak & Bocakova, 2008*).

## Male genitalia

The limits of most genera are currently based on the morphology of genitalia which is more reliable than external phenotypic characters. *Diatrichalus* has an exposed and complex internal sac (Figs. 5A–5C), *Lobatang* has a rod-shaped basal part of the internal sac (Figs. 5D, 5E and 5G–5I), *Eniclases* has the characteristic pigmented dorsal keel in the phallus (Fig. 5M) and *Flabellotrichalus* has the membranous, pigmented internal sac with a y-shaped basal part (Figs. 5N–5P). These characters were constant in respective genera and enable reliable identification, but they provide no information about deep relationships. Two sickle-like thorns at the base of the internal sac are present in *Trichalus* and *Microtrichalus* (Figs. 1B, 1C, 5F, 5J and 5K) and the preferred molecular phylogenetic hypothesis indicates their independent origin although with modest support (Fig. 1A). The presence of thorns in the internal sac is the principal character supporting their relationships in morphology-based analyses (Figs. 1B and 1C). Similar thorns are known in some *Synchonnus* (*Kusy, Sklenarova & Bocak, in press*; *Kusy, 2017*) and various members of distantly related genera of Metriorrhynchini, e.g., *Cautires* (*Jiruskova, Motyka & Bocak, 2016*).

## Female genitalia

The female genitalia provide additional information consistent with the molecular phylogenetic analyses. The strongest phenotypic character supporting the relationships among some trichaline genera are the dorsally attached lateral glands which define the clade (*Eniclases* + *Schizotrichalus*)((*Trichalus*, *Flabellotrichalus*) *Microtrichalus*). Other characters define the limits of genera, but do not contribute to the definition of more extensive clades. *Diatrichalus* has a characteristically long spermatheca (Fig. 6B) and all *Microtrichalus* have a pair of pockets in the middle part of the vagina and a slim unpaired ventral gland at the base of the vagina (Fig. 6H). With well-defined *Microtrichalus*, the genus *Trichalus* is left without any synapomorphy and its monophyly and relationships can be recovered only by molecular analyses (Fig. 1A).

## CONCLUSION

The phylogeny of the trichaline clade is separately recovered from morphology and molecular data, but neither analysis robustly solves all relationships. The deepest nodes in our phylogenies remain weakly supported by morphology, and only molecular analyses provide a stable topology with relatively high support for critical nodes (Figs. 1A–1C). The terminal clade of *Eniclases*, *Schizotrichalus*, *Trichalus*, *Flabellotrichalus,* and *Microtrichalus* is unambiguously supported by the unique morphology of vaginal glands, but only weakly so by the molecular data. The limits of all genera are congruently supported by morphological synapomorphies and molecular phylogenetic analyses, but

their robustness differs. *Diatrichalus* is well-delimited by several morphological characters but this clade receives only a low statistical support in our molecular analyses. The least supported genus-rank node is *Trichalus* (Fig. 1A), which is morphologically defined only by the absence of some phenotypic characters when compared with *Flabellotrichalus* and *Microtrichalus*. Similarly, this node obtains low statistical support in the molecular analyses (Fig. 1A).

The phenotypic characters can be misleading when similar structures evolve repeatedly or are so simplified that we are unable to identify homologues. Unexpectedly, the anterolateral pronotal carinae, lost in other trichaline genera, re-evolved in *Eniclases* and *Schizotrichalus*. Almost all trichaline species are unpalatable and aposematically colored, and due to their memberships in mimetic rings, the unrelated species can have similar body sizes and shapes (*Bocak & Yagi, 2010*). These homoplasious phenotypes attest to the strength of natural selection (*Bocek & Bocak, 2016*) and the traditionally used morphological characters, such as pronotal carinae, elytral costae and the shape of pronotum, display high intra-generic variability which might be caused by an independent origin of similar traits due to selective pressure. Further, the molecular phylogeny suggests repeated origins of flabellate antennae, which play a role in sexual communication. To summarize, the evaluation of both molecular and morphological signals is very valuable in net-winged beetles and their congruence should be evaluated whenever possible. Future studies can refine the trichaline classification, but a large part of the trichaline diversity has already been included in current analyses and we believe that the substantial rearrangements are improbable.

## ACKNOWLEDGEMENTS

We thank to the colleagues who provided an access to the type material in their care and the specimens for isolation of DNA: M. Barclay (London), K. Matsuda (Takarazuka City), R. Poggi (Genova), G. A. Samuelson (Honolulu), W. Schawaller (Stuttgart), S. A. Slipinski (Canberra), A. Taghavian (Paris), W. Tomaszewska (Warszawa). We are obliged to D. Richardson who critically red the manuscript before submission and to R. Bilkova for technical assistance.

### Funding

This work was supported by Czech Science Foundation, Czechia (Grant No: P506/11/1759), and by an IGA grant from the Faculty of Science UP, Czechia (Grant No: Prf-2017). The funders had no role in study design, data collection and analysis, decision to publish, or preparation of the manuscript.

### Grant Disclosures

The following grant information was disclosed by the authors:
Czech Science Foundation, Czechia: P506/11/1759.
Faculty of Science UP, Czechia: Prf-2017.

## Competing Interests

The authors declare there are no competing interests.

## Author Contributions

- Matej Bocek performed the experiments, analyzed the data, wrote the paper, prepared figures and/or tables, reviewed drafts of the paper.
- Ladislav Bocak conceived and designed the experiments, analyzed the data, contributed reagents/materials/analysis tools, wrote the paper, prepared figures and/or tables, reviewed drafts of the paper.

## DNA Deposition

The following information was supplied regarding the deposition of DNA sequences:

The sequences used here are accessible via GenBank accession numbers: MF288149–MF288557.

## Data Availability

The raw data has been supplied as Supplemental Dataset Files.

## New Species Registration

The following information was supplied regarding the registration of a newly described species:

Publication LSID: urn:lsid:zoobank.org:pub:BCDB57BC-DF3E-42A8-AB6D-2DCAB44799F3

Maibrius: urn:lsid:zoobank.org:act:0A2E45FB-72DB-49E7-BD7C-BC792072B106

Flabellotrichalus horaki: urn:lsid:zoobank.org:act:86069ACA-BC85-4865-847B-2EB421DC3BC3

## Supplemental Information

Supplemental information for this article can be found online at http://dx.doi.org/10.7717/peerj.3963#supplemental-information.

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
