# Peer review of "The comparison of molecular and morphology-based phylogenies of trichaline net-winged beetles (Coleoptera: Lycidae: Metriorrhynchini) with description of a new subgenus"

_PeerJ, doi:10.7717/peerj.3963_

## Round 0.1 · original submission · Minor Revisions

Dear Drs. Bocek and Bocak:
I am happy to report to you that two favorable reviews of your work have been conducted, and that the suggestions by both reviewers should greatly improve your manuscript. I foresee acceptance of your work upon addressing these reviews. In particular, please carefully consider the suggestions of Reviewer 1 regarding transparency of your methods, and remember that phylogenetic analyses should be 100% repeatable by a third party. This would entail including all model parameters (and reasons for choosing them) and making all character sets available to the public. Please also consider Reviewer 2's suggestion that outgroup selection needs to be justified.
Good luck with your revision, and I look forward to seeing your revised work!
-joe

Reviewer 1 ·

Basic reporting

The authors present a comprehensive molecular phylogeny of trichaline net-winged beetles whose taxonomic rank remains a subject of debate. The MS is written in clear English overall with minor editing (mostly grammatical) required to improve the coherence and sentence fluency (e.g., in line 131: “maximum parsimony (MP) analyses,” not “phylogenetic parsimony (MP) analyses;” and in line 136: “morphology-based phylogenetic relationships” rather than “phylogenetic morphology-based relationships”). Please also check for typos; there are still quite many, though I did not explicitly check for them. Sufficient review of relevant literature is provided and all figures and tables are prepared professionally.

Experimental design

The MS represents an original research that fits the aims and scope of the journal. The authors attempt to propose a natural classification system of trichaline net-winged beetles based on their comprehensive molecular phylogeny, together with morphological cladistic analyses at the genus level. The authors have employed a wide range of the state of the art methods with extensive sampling of molecular data. However, more detailed information and/or further elaboration on some of their analyses should be provided in Material and Methods.

- Lines 111–112: “The concatenated supermatrix was partitioned for all fragments and codon positions when appropriate.” Please provide how the final supermatrix was partitioned for each analysis. Was any software, such as PartitionFinder, used to determine the best partitioning scheme?

- Lines 113–116: Please explain why different models of nucleotide substitutions were used for ML tree search (i.e., GTRCAT) and its BS calculation (i.e., GTRGAMMA). In most cases, same model should be be used to calculate BS values.

- Lines 116–117: “The BI analysis was run in MrBayes . . . under the GTR+I+G model identified as above . . .” I highly recommend using softwares like PartitionFinder, which not only determines the best partition scheme, but also the best fitting models for each partition. The assignment of proper substitution model for each partition can significantly improve the phylogenetic reconstruction results.

- Explain how all the trees were rooted. This is essential for making assessment on monophyly of the ingroup, as well as its systematic position within the larger group (i.e., Metriorrhynchini).

Validity of the findings

The molecular phylogeny reported in the MS shows relationships among 143 samples representing 86 species of trichaline beetles, whereas the morphology-based phylogeny only include 11 terminals, each of which represents different genus within the so-called “trichaline clade.” The title of the MS, “The congruence of molecular and morphology-based phylogenies of trichaline net-winged beetles . . .,” can therefore be misleading since it sounds like the morphological analyses were conducted on the same set of samples included in molecular analyses.

Furthermore, the authors do not seem to have found strong evidence for the congruence between their DNA-based phylogeny and morphology-based phylogeny. In lines 266–271, the authors write: “The morphological analyses did not support the monophyly of the DNA-based trichaline clade. . . The deeper relationships were poorly supported,” and again in line 645: “The morphological analyses indicate different relationships,” which essentially refutes the main argument made on the congruence of molecular and morphology-based phylogenies. Therefore, a different title that more accurately describes the main finding of the MS seems necessary.

In lines 161–164: “The phylogenetic trees inferred . . . using the ML criterion and Bayesian inference were well-resolved and suggested similar relationships . . .” Please provide the BI topology as a supplementary figure and briefly discuss any discrepancy observed between the ML and BI trees, if there is any.

In lines 630–634: “The terminal position of the trichaline clade in Metriorrhynchina has already been demonstrated in the molecular analyses of Metriorrhynchini . . . Our analyses of the current more extensive dataset confirm the terminal placement of the trichaline clade (Fig. 1A).” This statement depends largely on how the trees were rooted in the present study. As I pointed out earlier, please explain in Material and Methods how your trees were rooted.

In the section Taxonomy, the diagnosis and redescription of “trichaline clade” is provided, but it is unclear which taxonomic rank this proposed clade represents. A clade can be a group of organisms at any taxonomic level, so long as the constituents are believed to be the decedents of a common ancestor. Therefore, the designation of trichaline as a “clade” does not address the main problem with alpha-taxonomy discussed in the introduction of the MS (see lines 42–43).

In Figure 1, note that there are two (B)’s in the legend and be mindful of their order (i.e., Fig.1B is the BI tree, and 1C the MP tree). Also, please clearly state that both 1B and 1C are morphology-based trees. I highly recommend collapsing all the nodes with support values below 50% in 1B and 1C trees and present them as polytomies, as already done for one of the nodes in Fig. 1C. You may provide all of the three original equally parsimonious MP trees as supplementary figures.

Reviewer 2 ·

Basic reporting

Manuscript is written in clear and concise English. References are ample, treated in concise format, sufficient background is provided in the intro part. Structure of the paper corresponds with usual arrangement of phylogenetic papers. Rich figures illustrates crucial structures of the morphological data, phylogenetic analyses are presented in clear and visually attractive form.

Molecular data are added as supplementary file in FASTA format. Morphological data are added as a table in the text. Probably it would be good to add it also as supplementary file, maybe in NEXUS file, to be quickly available for the readers?

Phylogenetic hypothesis is clearly formulated and correspond with text in results and discussion part of the ms.

Experimental design

The manuscript presents original biological research. Research questions are clearly formulated, corresponding with improvement in classification of the group, but also trying to link morphological and molecular data. This is highly interesting in a group developing mimetic rings, where many morphological characters have homoplastic distribution within different genera and classification based on morphology only can be highly misleading. This concept is clearly formulated in the text.

The research is rigorous, corresponds with high standards in phylogenetics, use advanced methods for evaluation of molecular data. In my opinion, it is in no conflicts with ethical standards.

Most of the methods are described in sufficient details. I have only two comments which can improve the clarity of the text:

1. Twenty species are treated as outgroup in the phylogenetic analysis. I will appreciate comments on logic how these taxa were selected – how closely related they are to the ingroup. Should be inserted in a paragraph on lines 107-122.

2. In morphological data matrix, the first taxon (Metriorrhynchus) presents probably outgroup and is used to root the resulting trees. This should be added to the text. Also, explicit information how character states were polarized should be mentioned in the text (paragraph on lines 124-142, 174-177).

List of morphological characters, included in Results (starting on line 174) should be moved in M&M section in my opinion. Some characters have detailed discussion about distribution of character states in individual genera or lineages, some have no comments. This should be unified. Several characters included presents only autapomorphies of a single genus, and bring no information to the phylogeny of genera (which is main goal of the paper).

Validity of the findings

I have no important comments to Results and Discussion parts, which are generally very well written. The following details can be however improved in the text:

Diagnosis of trichaline clade (lines 280-294). Here, on lines 286-294, you listed five very different states how male genitalia look like – this makes no diagnosis for the group! Maybe should be commented, or this part should be deleted?

Spinotrichalus, lines 427-428: “[these characters …] are the autapomorphy of S. telnovi and Spinotrichalis may be trated as synonym, when its position renders Lobatang paraphyletic.” But if you combine Spinotrichalus as subgenus of Lobatang, the same apply for the nominotypical subgenus!

The newly described subgenus, Maibrius, is characterized on lines 503-505 only as “differs from the nominotypical subgenus only in the serrate male antennae …” – but serrate/flabellate antennomeres are homoplasious within the group. You contradict this in the Discussion, lines 696-700), where you stated that “…serrate … male antennae … their value is questioned by variable morphology in related species”. Would it be possible to characterize Maibrius with some other morphological apomorphies as well?

Further, in a Remark on lines 507-513, you states that “the molecular phylogeny identified F. (Maibrius) horaki sp. n. as a genetically distant sister-lineage to other Flabellotrichalus”. But, as seen from Table 1 (lines 65-80), your sampling of this genus for molecular analysis is based exclusively on nine unidentified species (sp. A to sp. I in Table 1) from New Guinea. There is no info in this ms how many species of this genus is described, what is their geographical distribution etc. You should comment on this – why you are not able to identify your samples to species? Is your sampling covering the general distribution of known species of Flabellotrichalus – this means, is the nominotypical subgenus as proposed distributed in PNG and the newly described species from West Papua?

In the identification key, you are using exclusively male characters in some couples, but exclusively female ones in others (see lines 618-625). This is not very “user friendly”. Can this be improved? Also, info that female sex is not known for Maibrius should be inserted on lines 618-619.

Additional comments

Minor comments and typos:
line 50: “which is now a part” – it means a junior synonym or subgenus?
line 200: “1,” should be “(1)”
line 270: “Their stict consensus and one of the most parsimonious trees were unresolved” – I understand how consensus tree can have unresolved some nodes. But can be this applied also for a single parsimony three? Possible typo? Effect of missing data? This is not clear to me, please explain.
line 405: add comma
line 979: second “(B)” should be in fact “(C)”

---

## Round 0.2 · accepted · Accept

Dear Matej and Ladislav:

Thank you for submitting your revision to PeerJ. I am happy to report to you that your manuscript is now suitable for publication. Congratulations! Your attention to the reviewer's concerns has greatly improved the manuscript, and I believe this work will be an important contribution to the field. Well done!

Best,

-joe